# JURY-RL: VOTES PROPOSE, PROOFS DISPOSE FOR LABEL-FREE RLVR

## ABSTRACT

Reinforcement learning with verifiable rewards (RLVR) enhances the reasoning of large language models (LLMs), but its scalability is hampered by the high cost of human-annotated labels. Label-free alternatives, such as majority voting or LLM-as-a-judge, are susceptible to false positives that lead to reward hacking and training collapse. We introduce **JURY-RL**, a label-free RLVR framework that separates answer proposal from reward disposal: votes from model rollouts propose a consensus answer, while a formal theorem prover disposes the final reward. Specifically, a rollout is rewarded only if the majority-voted answer is formally verified by a Lean prover. When verification is inconclusive, we activate our proposed **ResZero (Residual-Zero)** reward: it drops the unverifiable majority proposal and assigns a zero-mean, variance-preserving reward to the remaining (residual) answers. This design maintains a stable optimization gradient for RL algorithms without reinforcing spurious consensus. Experiments across mathematical reasoning, code generation, and multi-task benchmarks show that JURY-RL consistently outperforms label-free baselines and attains performance comparable to supervised ground-truth training in pass@1, with superior generalization demonstrated by higher pass@k and response diversity.

## 1 INTRODUCTION

Large language models (LLMs) (OpenAI, 2023; DeepSeek-AI et al., 2024; Yang et al., 2025b) continue to advance in broad capabilities, yet reliable reasoning remains a core bottleneck (Wang et al., 2023; Lightman et al., 2024; Patil & Jadon, 2025; Huang et al., 2025b). Recent advances in reinforcement learning with verifiable rewards (RLVR) (Shao et al., 2024; Lambert et al., 2024; DeepSeek-AI et al., 2025) offers a principled post-training path: rather than aligning to what looks plausible, the objective aligns to what is provably correct by using verifiable signals such as from program execution or mathematical equivalence (Cobbe et al., 2021; Yu et al., 2025; AI & Agentica, 2025). However, scaling RLVR is constrained by its reliance on human-annotated answers or carefully curated specifications, which is costly and limited in coverage (Ouyang et al., 2022; Bai et al., 2022; Shao et al., 2024).

To reduce labeling cost, recent work explores label-free rewards, where no human ground-truth answers are provided during training. A prominent subset is self-supervised self-reward, which derives signals from the model or unlabeled data itself, including entropy minimization (Prabhudesai et al., 2025), self-certainty (Zhao et al., 2025b), and majority voting (Shafayat et al., 2025; Zhang et al., 2025). These approaches can learn reasoning but are prone to false positives and training collapse, often via reward hacking: models learn to satisfy the surrogate while drifting from correctness, (Gao et al., 2022; Shafayat et al., 2025). LLM-as-a-Judge (Lee et al., 2024; Su et al., 2025; Zhao et al., 2025c) provides another label-free path. Yet, in practice, they are vulnerable to instruction and format manipulation when reasoning steps are explicitly produced, and suffer high false positives when they are hidden. They are also sensitive to prompting and temperature, and introduce nontrivial compute overhead and shared-bias risks (Pang et al., 2023; Chen et al., 2024; Thakur et al., 2025; Shi et al., 2025; Zhao et al., 2025c;a).

The flaws of existing label-free methods, from reinforcing false consensus to reward hacking, are not isolated issues but rather symptoms of a deeper challenge. We argue that a truly robust reward signal must simultaneously satisfy three essential properties: (i) **scalable** without costly human

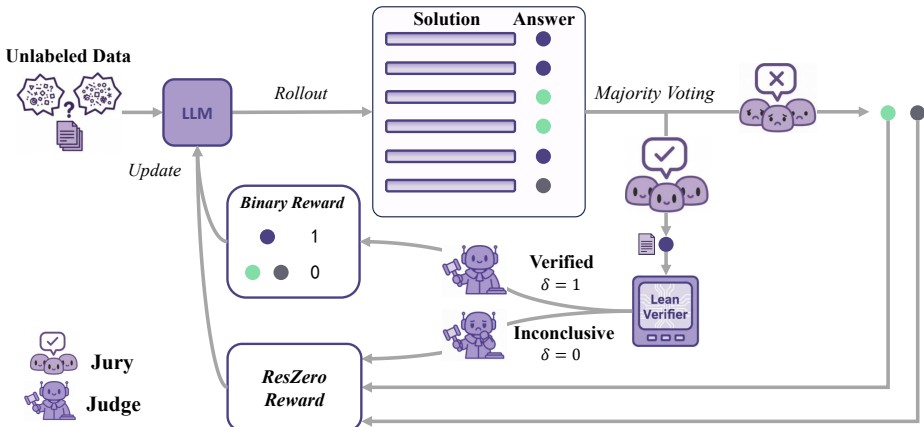

Figure 1: The **JURY-RL** workflow: Votes Propose, Proofs Dispose. For each problem, a majority vote from multiple rollouts (**Jury**) proposes a candidate answer. A lean verifier (**Judge**) then disposes the reward. If the answer is **Verified** ($\delta = 1$), supporting rollouts receive a positive reward, directly linking the learning signal to correctness. Conversely, when verification is **Inconclusive** ($\delta = 0$), all rollouts receive the proposed ResZero (Residual-Zero) Reward.

supervision, (ii) **truth-aligned**, rewarding verifiable correctness instead of error-prone consensus, and (iii) **optimization-stable**, allowing learning to proceed even when verification is inconclusive. Previous approaches have, in one degree or another, failed to meet all three criteria. Self-supervised signals such as majority voting achieve scalability but sacrifice truth-alignment, while LLM-as-a-Judge struggles with both truth-alignment due to high false positives and optimization instabilities.

This motivates our core proposal, a new paradigm designed to resolve this conflict: **Votes Propose, Proofs Dispose**. Our strategy is to decouple the *proposal* of a candidate answer from the final *disposal* of its reward. To maintain scalability, votes from model rollouts first propose a single consensus candidate through a computationally cheap majority vote [1]. A formal theorem prover (de Moura & Ullrich, 2021; Ren et al., 2025; Lin et al., 2025) then acts as a reliable judge to dispose the ultimate reward for this single candidate, thus satisfying all three principles above. This design choice avoids the prohibitive cost of formally verifying every unique answer, thereby making the entire framework viable at scale.

Our framework, **JURY-RL**, is shown in Figure 1 . A positive reward is granted only if the majority-voted answer is formally verified by a Lean verifier (de Moura & Ullrich, 2021), suppressing the false positives common in other self-supervised or judge-based methods (Shafayat et al., 2025; Gao et al., 2022). This raises a critical question: what happens when verification is inconclusive? A naive zero reward would stall learning, while rewarding the majority vote would reintroduce the risk of reinforcing errors. To solve this, we introduce the **ResZero (Residual-Zero)** reward, a novel fallback mechanism. ResZero discards the unverified majority proposal and assigns a carefully constructed, zero-mean reward to the remaining (residual) answers. This design maintains a stable optimization gradient for learning to proceed, without amplifying a potentially spurious consensus, ensuring both training stability and truth alignment.

**Contributions.** (1) We introduce **JURY-RL**, a novel label-free RLVR framework that operationalizes a "votes propose, proofs dispose" paradigm. By strategically verifying only the majority-voted candidate, it aligns rewards with provable correctness using a formal verifier, eliminating the need for human-annotated answers. (2) We design the **ResZero (Residual-Zero)** reward, a principled fallback mechanism for when verification is inconclusive. By discarding the unverifiable majority and assigning a zero-mean, variance-preserving reward to residual candidates, it ensures stable optimization and prevents collapse from spurious consensus. (3) Across mathematical reasoning, code generation, and multi-task benchmarks, JURY-RL trains more stably and achieves state-of-the-art results among label-free methods, matching a strong supervised ground-truth baseline in pass@1 while consistently surpassing it in pass@k and solution diversity.

---

[1]Throughout, "majority vote" follows the usual usage in the community and means a plurality.

## 2 RELATED WORK

**LLM Reasoning.** The general capabilities of LLMs have rapidly expanded (OpenAI, 2023; Dubey et al., 2024; Yang et al., 2025b), yet reliable mathematical and programmatic reasoning remains a bottleneck: models often optimize for plausibility rather than verifiable correctness (Ouyang et al., 2022; Rafailov et al., 2023; Touvron et al., 2023). Post-training techniques that elicit step-by-step reasoning (e.g., chain-of-thought and self-consistency) can raise average accuracy but also amplify confident but wrong results when no external check is available (Wei et al., 2022; Wang et al., 2023). These problems motivate recent verifiability-aligned training signals that reward what is provably correct rather than what appears correct (Shao et al., 2024; Lambert et al., 2024; Hu et al., 2025; DeepSeek-AI et al., 2025; Kimi-Team et al., 2025; Yang et al., 2025a).

**Label-Free RLVR.** To scale beyond labeled specifications, label-free surrogates derive rewards from the model or unlabeled data itself such as via majority voting (Shafayat et al., 2025; Zhang et al., 2025), confidence (Zhao et al., 2025b), entropy (Prabhudesai et al., 2025; Agarwal et al., 2025), or LLM-as-a-Judge (Pang et al., 2023; Lee et al., 2024; Su et al., 2025; Zhao et al., 2025a). While attractive for its broad coverage and low cost, these signals are prone to false positives, prompt/format gaming (Zhao et al., 2025c), and training collapse (Zhang et al., 2025). As a result, consensus or judge approval models risk diverging from ground truth, leading to reward hacking and instability (Shafayat et al., 2025; Zhang et al., 2025). Our work targets this conflict: retain the scalability of label-free training while removing optimism toward unverified agreement.

**Lean and Other Verifiers.** Verification-based training employs externally checkable signals such as program execution and unit tests, SMT solvers, or formal proof assistants such as Lean/Coq to couple reward with correctness (P.Huet et al., 1997; C.Blanchette et al., 2011; de Moura & Ullrich, 2021; Cobbe et al., 2021; AI & Agentica, 2025). Previous generate-then-verify pipelines typically provide no learning signal when verification fails, limiting stability and sample efficiency. JURY-RL decouples proposal from disposal to preserve label-free scalability while maintaining optimization alignment with provable correctness. Crucially, unlike process reward approaches or hybrid verifiers (Huang et al., 2025c) that rely on learned judges, JURY-RL employs the Lean proof kernel as the sole ground-truth oracle. We treat the autoformalization pipeline strictly as a generation mechanism rather than a supervision source, ensuring that positive rewards are gated exclusively by formal verification reliability rather than susceptible learned approximations.

## 3 PRELIMINARIES

**Problem Setup.** Let $\pi_\theta$ denote a policy LLM with parameters $\theta$. Given a problem $x$, the model generates a token sequence $y = (y_1, \ldots, y_n) \sim \pi_\theta(\cdot \mid x)$ and a deterministic extractor $\mathrm{ans}(\cdot)$ parses a candidate answer $a = \mathrm{ans}(y)$. In the *label-free* setting, ground-truth answers are unavailable during training. Instead, each $x$ can be associated with a machine-checkable specification $\mathrm{spec}(x)$, and an external verifier (e.g., a Lean-based checker) exposes a binary oracle

$$\mathrm{verify}(x, a) \in \{0, 1\},$$

which returns 1 if $a$ is *formally* certified correct under standard soundness assumptions.

We optimize a KL-regularized RLVR objective with a reference policy $\pi_{\mathrm{ref}}$ and coefficient $\beta$:

$$\max_{\pi_\theta} \ \mathbb{E}_{x \sim \mathcal{D}, \, y \sim \pi_\theta(\cdot|x)} \Big[ r(x, y; \mathcal{G}_x) \ - \ \beta \, \mathrm{D}_{\mathrm{KL}}\big(\pi_\theta(\cdot \mid x) \,\|\, \pi_{\mathrm{ref}}(\cdot \mid x)\big) \Big], \tag{1}$$

where $r(\cdot)$ is a *grouped* reward computed from $G$ rollouts $\mathcal{G}_x = \{y_i\}_{i=1}^G$ for the same input.

We adopt Group Relative Policy Optimization (GRPO) (Shao et al., 2024) to estimate group-normalized advantages. Concretely, sample $y_i \sim \pi_{\mathrm{old}}(\cdot \mid x)$ for $i = 1, \ldots, G$ and compute $r_i := r(x, y_i; \mathcal{G}_x)$. The scalar group advantage is

$$\hat{A}_i \ = \ \frac{r_i - \bar{r}}{\mathrm{std}(\{r_j\}_{j=1}^G) + \varepsilon}, \qquad \bar{r} = \tfrac{1}{G} \sum_{j=1}^G r_j. \tag{2}$$

Let the per-token ratio be $\rho_{i,t}(\theta) = \frac{\pi_\theta(y_{i,t}|x,y_{i,<t})}{\pi_{\text{old}}(y_{i,t}|x,y_{i,<t})}$. GRPO maximizes

$$J_{\text{GRPO}}(\theta) \;=\; \mathbb{E}\!\left[\frac{1}{G}\sum_{i=1}^{G}\frac{1}{|y_i|}\sum_{t=1}^{|y_i|}\min\!\big(\rho_{i,t}(\theta)\hat{A}_{i,t},\,\text{clip}(\rho_{i,t}(\theta),1-\epsilon,1+\epsilon)\hat{A}_{i,t}\big) - \beta\, D_{\text{KL}}(\pi_\theta\|\pi_{\text{ref}})\right],$$

(3)

where $\hat{A}_{i,t}$ is a broadcast of $\hat{A}_i$ (or any per-token variant compatible with GRPO), and the clipping avoids excessive policy drift. This presentation mirrors established RLVR practice for fair comparison and faithful reproduction.

**Label-free Self-Reward in Reinforcement Learning.** We categorize existing methods for label-free self-reward based on the origin of the reward signal: model-internal signals versus those from an external judge.

*(A) Model-Internal Self-Reward Signals.* These methods derive rewards from the model's own outputs without external evaluators. (i) **Output-side proxies**, such as entropy minimization or confidence-based scores (Agarwal et al., 2025; Prabhudesai et al., 2025), reward hypotheses that exhibit high certainty. However, this approach is fragile, as it can amplify errors when the model becomes confidently wrong. (ii) **Single-view agreement** rewards consistency among multiple outputs generated from the same input $x$ (Shafayat et al., 2025). Specifically, for $G$ responses $y_i{}_{i=1}^{G}$, it identifies the majority-voted answer $a_v = \arg\max_a \sum_{i=1}^{G} \mathbb{I}[\text{ans}(y_i) = a]$ and assigns a positive reward $r_{\text{sv}}(x,y) = \mathbb{I}[\text{ans}(y) = a_v]$ to responses that match $a_v$. The primary risk is reinforcing an erroneous consensus, where the model converges on a popular but incorrect answer, often by exploiting superficial heuristics such as formatting conventions. (iii) **Multi-view agreement** attempts to improve robustness by enforcing consistency across multiple, semantically equivalent prompts. For instance, the majority answer from prompt variant $x'$ is used as a pseudo-label to supervise responses from the original prompt $x$ (Zhang et al., 2025). This often improves training stability. However, it usually only delays rather than eliminates hacking, since spurious shortcuts can eventually propagate across multiple views.

*(B) External-Judge Signals.* This paradigm uses a powerful, external LLM as an automated judge to score the model's outputs. **LLM-as-a-Judge** (Lee et al., 2024; Su et al., 2025; Huang et al., 2025a; Zhao et al., 2025a) remains label-free (no human annotation) but comes with distinct trade-offs. On the one hand, it mitigates the self-confirmation bias of internal methods. On the other hand, it introduces sensitivity to the judge's prompt design and decoding strategy, incurs significant computational cost, and risks transferring the judge's intrinsic biases into the training signal. Another approach is **LLM-based Knowledge Distillation (LLM-KD)** (Gu et al., 2024b), where a teacher model generates a reference answer to guide the student model. While potentially offering a more granular signal, it is similarly constrained by the teacher's capabilities and biases. Thus, while external judges reduce some weaknesses of the internal proxies, they bring a different set of reliability and scalability challenges.

**Positioning Our Work.** JURY-RL is a label-free RLVR method that *proof-gates* reward: a majority-voted answer is rewarded only if a formal verifier certifies it. When verification is inconclusive, we drop the majority and apply **ResZero**—a zero-mean, variance-preserving residual reward, so that GRPO maintains a stable gradient without reinforcing spurious consensus. This contrasts with the self-reward approaches that reward popularity or confidence, and with LLM-as-a-Judge/KD approaches that are prompt-sensitive and prone to false positives; JURY-RL aligns learning to verifiable correctness while remaining label-free.

## 4 JURY-RL

JURY-RL is designed to satisfy three core principles: (i) **scalability** without costly human supervision, (ii) **truth alignment** by grounding rewards in verifiable evidence, and (iii) **optimization stability**, ensuring continuous learning even when verification is inconclusive. We achieve this by decoupling the process of proposing a candidate answer from the final disposal of its reward. Votes from the policy's own rollouts serve as a scalable proposal mechanism, while a formal theorem prover acts as a reliable judge for reward disposal. This design choice is crucial for maintaining computational tractability and scalability. Performing formal verification on every unique answer

generated across all rollouts would be prohibitively expensive, undermining the efficiency of the label-free approach. The majority vote thus acts as an effective heuristic to identify the most promising candidate for the costly-but-reliable verification process.

This section details the two key components of our framework: the overarching **proof-gated reward mechanism** that enforces truth alignment, and the **Residual-Zero (ResZero) fallback** designed to maintain optimization stability.

## 4.1 THE PROOF-GATED REWARD FRAMEWORK

The JURY-RL workflow begins with a **proposal stage**. For a given problem $x$, we generate $G$ trajectories $\{y_i\}_{i=1}^G \sim \pi_\theta(\cdot|x)$ and parse their corresponding answers $a_i = \mathrm{ans}(y_i)$. A majority vote determines the most frequent answer, which becomes our candidate proposal:

$$\hat{a} = \arg \max_a \sum_{i=1}^G \mathbb{I}[a_i = a]$$

This proposal is then passed to the **disposal stage**. A single call to an external Lean verifier [2] evaluates the correctness of $\hat{a}$ against a formal specification of $x$. This yields a binary *proof-gate*, $\delta = \mathrm{verify}(x, \hat{a}) \in \{0, 1\}$, which dictates the reward assignment.

The final reward $r_i$ for each trajectory $y_i$ is determined by a conditional function gated by $\delta$:

$$r_i = \underbrace{\delta \cdot \mathbb{I}[a_i = \hat{a}]}_{\text{Verified Correctness}} + \underbrace{(1 - \delta) \cdot r_i^{\text{ResZero}}}_{\text{Inconclusive Fallback}}, \tag{4}$$

where $r_i^{\text{ResZero}}$ is the Residual-Zero Reward detailed in Section 4.2.

The stability of this proof-gated design stems from its principled handling of both successful and inconclusive verification. First, when verification succeeds ($\delta = 1$), a positive reward is granted exclusively to the trajectories that produced the proven-correct answer. This approach directly binds the learning signal to hard evidence, which, under standard soundness assumptions, suppresses the false positives that plague self-reward or judge-based surrogates (Zhang et al., 2025; Cobbe et al., 2021). Second, when verification is inconclusive ($\delta = 0$), the system defaults to our ResZero fallback rather than rewarding the majority consensus. This centered substitute maintains a stable optimization gradient by preserving group-wise variance, which prevents learning from stalling or oscillating, a common situation when verification fails for reasons like search limits or incomplete proofs. Finally, by paying only for verifiable correctness, this framework inherently narrows the attack surface for prompt and format hacking, a critical vulnerability in other label-free systems.

## 4.2 RESZERO REWARD

When formal verification is inconclusive, a learning signal is still needed to maintain optimization stability. However, a naive choice like directly rewarding the majority vote (MV) is brittle and risks training collapse. Using MV as a reward conflates agreement with correctness and can induce entropy collapse under GRPO, as spurious consensus strengthens. A simple zero-reward fallback is also suboptimal, as it would lead to zero group-wise advantage and stall the learning process.

To address this, we introduce the **ResZero (Residual-Zero) Reward**. Its principle is to **penalize the unverifiable majority proposal and construct a meaningful, zero-mean reward from the remaining (residual) answers.** This design preserves a useful learning signal by maintaining variance among minority opinions without amplifying a potentially false consensus. Furthermore, we propose an adaptive variant that strengthens this signal precisely when the model is most confidently wrong. The intuition is that a strong but unverified consensus (indicated by a high majority share, $\alpha$) requires a stronger corrective signal. ResZero operationalizes this by using $\alpha$ to simultaneously amplify the reward signal for residual answers and suppress the majority answer.

Let $M = \{i : a_i = \hat{a}\}$ be the set of rollouts supporting the majority answer and $R = \{i : a_i \neq \hat{a}\}$ be the set of residual rollouts. The majority share is $\alpha = |M|/G$. We first define the leave-one-out

---

[2]Details of our Lean verifier can be found in Appendix C.

residual share for an answer $b$ within the residual group:

$$u^{(-i)}(b) = \frac{1}{|R|-1} \sum_{\substack{j \in R \\ j \neq i}} \mathbb{I}[a_j = b].$$

Let $z_i = u^{(-i)}(a_i)$ if $i \in R$ (the relative support for a residual answer $a_i$ within its peer group) and $z_i = 0$ if $i \in M$. The **ResZero** reward is then assigned as:

$$r_i^{\text{ResZero}} = \underbrace{\alpha \cdot \mathbb{I}[i \in R] \cdot (z_i - \bar{u})}_{\text{Amplify residual signals}} - \underbrace{c\alpha \cdot \mathbb{I}[i \in M]}_{\text{Penalize majority}} + \underbrace{\gamma}_{\text{Global re-centering}},$$

$$\text{where} \quad \bar{u} = \frac{1}{|R|} \sum_{j \in R} z_j, \quad \text{and} \quad \gamma = c\alpha^2. \tag{5}$$

Here, $c$ is a positive hyperparameter controlling the penalty strength. $\gamma$ is a global re-centering term to ensure the total reward sums to zero. The term $(z_i - \bar{u})$ creates a zero-mean signal *within the residual group*, rewarding answers that are more popular among the minorities and penalizing those that are less so. This entire residual signal is then scaled by $\alpha$. By construction, the total reward sums to zero ($\sum_i r_i^{\text{ResZero}} = 0$), preserving the zero-mean property crucial for GRPO stability. [3]

The design of our ResZero reward ensures robust optimization through three properties. (i) **Variance preservation**: It maintains non-zero variance among differing answers, which is critical for variance-normalized optimizers like GRPO to prevent vanishing gradients. (ii) **Zero-mean construction**: Its strictly zero-mean property makes it a principled, optimizer-agnostic signal, ensuring portability beyond GRPO to any general RL paradigm. (iii) **Adaptive economy**: The corrective signal dynamically scales with the majority share $\alpha$, applying maximum pressure when the model is most confidently wrong, with this entire behavior governed by only a single hyperparameter $c$. These properties prevent collapse into spurious consensus and foster robust, exploratory learning.

This dual-reward strategy hinges on the verifier's outcome. If verification succeeds ($\delta=1$), the policy update is guided by verifiable correctness. Conversely, if inconclusive ($\delta=0$), our ResZero fallback penalizes the unverified majority, which is critical for mitigating entropy collapse caused by spurious consensus. This design maintains the scalability of label-free training by requiring only a single verification per step. The full procedure is detailed in Appendix B.

## 5 EXPERIMENTS

### 5.1 SETTING

**Backbone Models.** Our experiments are conducted on a diverse range of open-source large language models to ensure broad applicability. This includes models from the Qwen2.5 (Yang et al., 2025b), Qwen3 series (Yang et al., 2025a), and the Llama3 series (Dubey et al., 2024).

**Baselines.** We compare JURY-RL with several established label-free and supervised reward baselines. The self-supervised baselines include **Majority-Voting** (Shafayat et al., 2025), **Self-Certainty** (Zhao et al., 2025b), **Entropy** minimization (Prabhudesai et al., 2025) and **CoReward** (Zhang et al., 2025). To further contextualize the comparison, we include **ground-truth supervised oracle** and **LLM-KD** as baselines. Additionally, we benchmark against judge-based method, **LLM-as-a-Judge** (Pang et al., 2023; Zhao et al., 2025a), to provide a comprehensive evaluation. Additional details are provided in Appendix D.

**Implementation Details.** All methods are implemented using the VeRL framework (Sheng et al., 2025) and trained on $8\times$ NVIDIA A100 GPUs. We train on 7,500 problems from the MATH dataset's training split (Hendrycks et al., 2021), and evaluate on the 5,000-problem validation split (referred to as MATH5000). For each GRPO (Shao et al., 2024) update step, we sample a batch of 128 problems and generate $G = 8$ rollouts per problem. We use a learning rate of $3 \times 10^{-6}$ and a KL penalty coefficient of $\beta = 0.005$. To ensure fair and reproducible comparisons, we utilize the officially released chat-based prompting formats for all models. More details in Appendix E.

---

[3]See App. A.2 for a formal proof and App. B.2 for a worked example. $c = 0.01$ in our experiments.

Table 1: RL performance (%) on math reasoning benchmarks. Cell background colors indicate relative performance: darker colors denote better results within each model group.

| Methods | Mathematics | | | | | Code | | Instruction | Multi-Task | Average |
|---------|-------|-------|----------|-------|-----|----------|------|--------|----------|---------|
| | AIME24 | AIME25 | MATH-500 | GSM8K | AMC | LiveCode | CRUX | IFEval | MMLU-Pro | |
| *Qwen3-1.7B-Base* | | | | | | | | | | |
| Before RL | $3.12_{\pm1.8}$ | $1.25_{\pm1.1}$ | $46.85_{\pm2.9}$ | $64.20_{\pm1.8}$ | $20.78_{\pm2.5}$ | $4.59_{\pm0.7}$ | $7.52_{\pm1.1}$ | $33.66_{\pm1.6}$ | $33.60_{\pm0.7}$ | 23.95 |
| GT-Reward | $6.88_{\pm2.1}$ | $5.21_{\pm2.0}$ | $68.35_{\pm2.3}$ | $82.01_{\pm1.1}$ | $30.87_{\pm3.3}$ | $14.84_{\pm0.5}$ | $33.73_{\pm0.6}$ | $38.70_{\pm1.6}$ | $39.43_{\pm0.7}$ | 35.56 |
| LLM-KD | $6.67_{\pm1.1}$ | $3.54_{\pm1.2}$ | $67.8_{\pm1.2}$ | $81.97_{\pm1.3}$ | $35.69_{\pm2.2}$ | $14.05_{\pm0.9}$ | $33.80_{\pm1.5}$ | $34.65_{\pm1.6}$ | $37.97_{\pm0.7}$ | 35.13 |
| Self-Certainty | $3.54_{\pm1.6}$ | $3.12_{\pm1.8}$ | $57.05_{\pm2.7}$ | $73.71_{\pm1.4}$ | $27.41_{\pm2.3}$ | $10.37_{\pm0.9}$ | $18.73_{\pm1.3}$ | $38.58_{\pm1.6}$ | $35.28_{\pm0.7}$ | 29.75 |
| Entropy | $6.67_{\pm1.6}$ | $4.17_{\pm1.5}$ | $65.0_{\pm1.7}$ | $79.45_{\pm1.6}$ | $31.17_{\pm3.0}$ | $12.64_{\pm1.0}$ | $30.38_{\pm0.5}$ | $35.00_{\pm1.6}$ | $37.01_{\pm0.7}$ | 33.50 |
| Majority-Voting | $2.08_{\pm1.3}$ | $2.08_{\pm1.3}$ | $59.6_{\pm0.9}$ | $81.08_{\pm0.8}$ | $29.82_{\pm1.7}$ | $14.29_{\pm0.5}$ | $32.00_{\pm2.3}$ | $37.16_{\pm1.6}$ | $35.68_{\pm0.7}$ | 32.64 |
| CoReward | $6.88_{\pm1.5}$ | $3.12_{\pm1.5}$ | $65.4_{\pm1.2}$ | $81.07_{\pm0.8}$ | $33.13_{\pm3.0}$ | $14.27_{\pm0.5}$ | $34.10_{\pm1.1}$ | $37.28_{\pm1.6}$ | $37.39_{\pm0.7}$ | 34.74 |
| LLM-as-a-Judge | $5.83_{\pm1.5}$ | $1.88_{\pm0.9}$ | $63.0_{\pm2.4}$ | $81.84_{\pm1.4}$ | $33.89_{\pm1.1}$ | $14.33_{\pm0.8}$ | $34.98_{\pm1.5}$ | $36.28_{\pm1.6}$ | $33.83_{\pm0.7}$ | 33.98 |
| **JURY-RL (Ours)** | $7.71_{\pm1.8}$ | $4.58_{\pm2.0}$ | $66.4_{\pm2.0}$ | $82.58_{\pm0.3}$ | $34.19_{\pm3.1}$ | $14.54_{\pm0.4}$ | $33.98_{\pm2.5}$ | $36.55_{\pm1.6}$ | $38.11_{\pm0.7}$ | 35.40 |
| *Llama-3.2-3B-Instruct* | | | | | | | | | | |
| Before RL | $6.88_{\pm2.1}$ | $0.42_{\pm0.6}$ | $44.4_{\pm2.1}$ | $68.78_{\pm1.2}$ | $17.47_{\pm3.3}$ | $3.00_{\pm0.4}$ | $24.70_{\pm1.2}$ | $54.41_{\pm1.8}$ | $32.01_{\pm0.7}$ | 28.01 |
| GT-Reward | $10.62_{\pm2.1}$ | $0.21_{\pm0.4}$ | $47.5_{\pm1.4}$ | $78.60_{\pm0.8}$ | $22.89_{\pm1.9}$ | $7.05_{\pm0.6}$ | $32.48_{\pm0.8}$ | $50.16_{\pm1.7}$ | $34.26_{\pm0.7}$ | 31.53 |
| LLM-KD | $10.21_{\pm1.9}$ | $0.42_{\pm0.6}$ | $51.25_{\pm4.5}$ | $79.72_{\pm1.2}$ | $24.55_{\pm3.4}$ | $7.55_{\pm0.6}$ | $31.20_{\pm1.6}$ | $49.06_{\pm1.7}$ | $34.04_{\pm0.7}$ | 32.00 |
| Self-Certainty | $2.71_{\pm1.3}$ | $0.62_{\pm0.7}$ | $42.4_{\pm0.9}$ | $73.52_{\pm1.2}$ | $17.62_{\pm1.8}$ | $5.57_{\pm0.9}$ | $24.55_{\pm1.2}$ | $54.11_{\pm1.8}$ | $34.42_{\pm0.7}$ | 28.39 |
| Entropy | $3.54_{\pm1.5}$ | $0.21_{\pm0.4}$ | $40.5_{\pm2.9}$ | $68.57_{\pm1.7}$ | $17.17_{\pm2.6}$ | $5.38_{\pm0.8}$ | $26.30_{\pm1.7}$ | $54.24_{\pm1.8}$ | $33.54_{\pm0.7}$ | 27.72 |
| Majority-Voting | $8.33_{\pm1.4}$ | $0.00_{\pm0.0}$ | $48.4_{\pm3.0}$ | $78.64_{\pm1.4}$ | $21.69_{\pm2.8}$ | $8.11_{\pm0.5}$ | $30.83_{\pm1.4}$ | $48.36_{\pm1.7}$ | $33.94_{\pm0.7}$ | 30.92 |
| CoReward | $7.71_{\pm1.4}$ | $0.00_{\pm0.0}$ | $47.2_{\pm1.9}$ | $80.00_{\pm1.7}$ | $19.58_{\pm3.6}$ | $5.57_{\pm0.4}$ | $30.87_{\pm1.2}$ | $50.46_{\pm1.7}$ | $32.95_{\pm0.7}$ | 30.48 |
| LLM-as-a-Judge | $9.38_{\pm1.9}$ | $0.62_{\pm0.7}$ | $47.35_{\pm2.5}$ | $78.51_{\pm2.2}$ | $20.93_{\pm3.4}$ | $3.96_{\pm0.5}$ | $31.45_{\pm1.4}$ | $51.46_{\pm1.7}$ | $34.22_{\pm0.7}$ | 30.88 |
| **JURY-RL (Ours)** | $9.17_{\pm2.0}$ | $1.25_{\pm0.9}$ | $48.8_{\pm2.6}$ | $78.96_{\pm0.5}$ | $26.05_{\pm1.6}$ | $6.16_{\pm0.6}$ | $31.60_{\pm1.2}$ | $50.09_{\pm1.8}$ | $34.54_{\pm0.7}$ | 31.85 |
| *Qwen2.5-7B* | | | | | | | | | | |
| Before RL | $4.38_{\pm1.4}$ | $1.67_{\pm1.4}$ | $49.2_{\pm2.6}$ | $71.00_{\pm1.9}$ | $22.44_{\pm3.7}$ | $4.57_{\pm0.5}$ | $27.38_{\pm2.3}$ | $40.61_{\pm1.7}$ | $44.03_{\pm0.8}$ | 29.48 |
| GT-Reward | $18.33_{\pm2.3}$ | $11.25_{\pm1.7}$ | $75.1_{\pm1.4}$ | $90.83_{\pm0.5}$ | $46.54_{\pm2.5}$ | $12.78_{\pm1.6}$ | $53.67_{\pm1.0}$ | $41.50_{\pm1.7}$ | $45.09_{\pm0.7}$ | 43.90 |
| LLM-KD | $12.08_{\pm2.0}$ | $7.92_{\pm1.7}$ | $72.7_{\pm1.2}$ | $90.96_{\pm0.9}$ | $44.28_{\pm2.1}$ | $8.09_{\pm0.5}$ | $53.57_{\pm0.5}$ | $43.09_{\pm1.7}$ | $43.73_{\pm0.7}$ | 41.82 |
| Self-Certainty | $9.79_{\pm2.4}$ | $8.96_{\pm1.8}$ | $72.45_{\pm2.8}$ | $88.21_{\pm0.5}$ | $40.51_{\pm2.1}$ | $11.87_{\pm0.9}$ | $53.18_{\pm1.4}$ | $39.66_{\pm1.7}$ | $43.89_{\pm0.7}$ | 40.95 |
| Entropy | $11.04_{\pm2.3}$ | $8.33_{\pm2.2}$ | $73.15_{\pm1.7}$ | $87.85_{\pm0.6}$ | $40.51_{\pm3.1}$ | $15.68_{\pm0.9}$ | $51.08_{\pm1.3}$ | $40.25_{\pm1.7}$ | $42.61_{\pm0.7}$ | 41.17 |
| Majority-Voting | $11.25_{\pm2.1}$ | $4.17_{\pm1.2}$ | $71.0_{\pm0.7}$ | $90.52_{\pm0.7}$ | $38.70_{\pm1.3}$ | $18.37_{\pm0.9}$ | $52.20_{\pm1.1}$ | $42.72_{\pm1.7}$ | $43.83_{\pm0.8}$ | 41.42 |
| CoReward | $11.25_{\pm1.8}$ | $3.96_{\pm1.9}$ | $73.85_{\pm1.1}$ | $90.16_{\pm0.9}$ | $40.81_{\pm1.8}$ | $10.37_{\pm0.5}$ | $55.08_{\pm2.4}$ | $43.75_{\pm1.7}$ | $42.08_{\pm0.7}$ | 41.26 |
| LLM-as-a-Judge | $10.42_{\pm2.1}$ | $5.62_{\pm1.2}$ | $72.3_{\pm1.2}$ | $89.73_{\pm0.7}$ | $41.11_{\pm3.1}$ | $10.58_{\pm0.3}$ | $51.90_{\pm1.6}$ | $44.17_{\pm1.7}$ | $48.67_{\pm0.7}$ | 41.61 |
| **JURY-RL (Ours)** | $13.75_{\pm1.8}$ | $7.50_{\pm1.6}$ | $75.65_{\pm0.7}$ | $90.35_{\pm0.9}$ | $47.29_{\pm3.2}$ | $14.69_{\pm1.1}$ | $54.37_{\pm0.4}$ | $41.51_{\pm1.7}$ | $50.00_{\pm0.8}$ | 43.90 |

**Evaluation Datasets and Metrics.** To comprehensively assess model capabilities, we evaluate on a suite of benchmarks covering mathematical reasoning, code generation, and general abilities. **Mathematical Reasoning:** We evaluate on the AIME24/25 (Hugging Face H4, 2024; OpenCompass, 2025), MATH500 (Lightman et al., 2024), GSM8K (Cobbe et al., 2021), and the competition-level AMC datasets (math-ai Team, 2024). **Code Generation:** We assess coding proficiency using LiveCodeBench (Jain et al., 2024) and CRUX (Gu et al., 2024a). **Instruction-Following and Multi-Task:** General abilities are measured using IFEval (Zhou et al., 2023) for instruction following and MMLU-Pro (Wang et al., 2024) for multi-task understanding. The specific metrics, frameworks, and implementation details are provided in Appendix F.

## 5.2 MAIN RESULTS

We evaluate **JURY-RL** on both in- and out-domain tasks, as shown in Tables 1 under avg@k settings[4]. A key finding is that JURY-RL not only outperforms all label-free RL baselines but also consistently matches the supervised GRPO with ground-truth rewards (GT), suggesting that proof-gated rewards can offer a promising learning signal to direct supervision.

**Mathematical Reasoning and In-Domain Generalization.** The results on mathematical benchmarks in Table 1 reveal a crucial insight into JURY-RL's learning mechanism. We observe that while its advantage over baselines is modest on the **in-distribution** MATH500 test set (which shares the same domain as our MATH training data), its superiority becomes substantially more pronounced on **out-of-distribution** math benchmarks such as GSM8K and AMC. We posit that this pattern arises because competing methods, including the supervised GT baseline, tend to overfit to the stylistic patterns and problem-solving shortcuts of the MATH dataset. In contrast, JURY-RL, by relying on formal verification, is incentivized to learn the underlying mathematical principles that are robust to such distributional shifts. This superior generalization within **the math domain** yields math performance that is comparable to the supervised GT baseline across all three backbones. In par-

---

[4]Values are reported as mean $\pm$ 95% confidence intervals.

ticular, JURY-RL improves average accuracy on challenging out-of-distribution benchmarks such as GSM8K and AMC, while maintaining similar performance on the in-distribution MATH500 set. This body of evidence indicates that the signal from formal verification is not merely a proxy for ground truth but can be a more potent objective for learning generalizable reasoning.

**Out-of-Domain Generalization.** The model exhibits robust out-of-domain performance across code generation, instruction following, and multi-task knowledge benchmarks. On Qwen2.5-7B, JURY-RL again surpasses the GT baseline on these out-of-domain benchmarks, achieving an average of **40.14%** (+1.88 pts, +4.92% rel.). On the other two backbone models, its performance remains close to GT while consistently ranking as the best-performing label-free method. These results demonstrate that optimizing for verifiable correctness encourages the model to learn fundamental, transferable skills that generalize beyond the mathematical domain used for training.

**Overall Gains.** Across all three backbones and different benchmarks, **JURY-RL** consistently delivers the strongest overall performance among label-free RL methods. In terms of the aggregated averages in Table 1, it uniformly outperforms other label-free methods with especially clear gains on challenging out-of-distribution math and code benchmarks. At the same time, its overall performance remains competitive with supervised variants like GT and LLM-KD. JURY-RL closely tracks or slightly surpasses these supervised baselines across backbones, while avoiding the training instabilities and collapse behaviors observed in other label-free approaches. Taken together, these results show that optimizing for verifiable correctness yields a label-free objective that is both more effective than prior self-supervised methods and sufficiently strong to serve as a promising way to scale up label-free RLVR.

### 5.3 ANALYSIS

**Does JURY-RL Enhance Diversity?** Yes. Table 2 shows that JURY-RL improves both pass@k and pass@1 over GT-Reward and LLM-as-a-judge on all three backbones. On average, the gains in pass@k under multi-sample decoding are substantially larger than those in pass@1, especially compared to LLM-as-a-judge (e.g., +9.06 and +10.05 percentage points in pass@k on Qwen3-1.7B-Base and Qwen2.5-7B). This pattern indicates that JURY-RL increases the number of distinct high-quality solutions rather than only refining the single best prediction. Our **ResZero** reward contributes to this behavior by down-weighting unverifiable majority answers and redistributing reward to residual trajectories, which promotes exploration of alternative reasoning paths and mitigates mode collapse. Consistently, Figure 2 shows that the average number

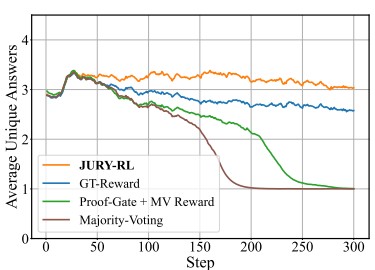

Figure 2: Average unique answers per sample over training steps on Qwen3-1.7B-Base.

of unique answers per problem remains high throughout training for JURY-RL, whereas baselines such as Majority-Voting quickly collapse to a single dominant answer. Per-benchmark pass@k breakdowns are provided in Table 8, Appendix G.1.

Table 2: Performance of JURY-RL vs. GT-Reward and LLM-as-a-judge on math reasoning tasks. k=16 for AIME, k=4 for MATH500 and GSM8K, and k=8 for AMC. $\Delta_{GT}$ and $\Delta_{LJ}$ denote JURY-RL's improvements (in percentage points) over GT-Reward and LLM-as-a-judge, respectively.

| Model | Average (pass@k) | | | | | Average (pass@1) | | | | |
|---|---|---|---|---|---|---|---|---|---|---|
| | GT-Reward | LLM-as-a-judge | JURY-RL | $\Delta_{GT}$ (pp) | $\Delta_{LJ}$ (pp) | GT-Reward | LLM-as-a-judge | JURY-RL | $\Delta_{GT}$ (pp) | $\Delta_{LJ}$ (pp) |
| Qwen3-1.7B-Base | 55.36 | 50.35 | **59.41** | +4.05 | +9.06 | 39.25 | 37.60 | **41.57** | +2.32 | +3.97 |
| Llama-3.2-3B-Instruct | 45.46 | 45.93 | **48.48** | +3.02 | +2.55 | 32.19 | 30.65 | **34.10** | +1.91 | +3.45 |
| Qwen2.5-7B | 62.48 | 53.99 | **64.04** | +1.56 | +10.05 | 46.60 | 44.13 | **48.13** | +1.53 | +4.00 |

**Ablation Studies of ResZero.** We compare four fallback designs under the same proof-gated framework when $\delta=0$: Zero Reward (no signal), Random Reward (rewarding a randomly selected candidate), MV Reward, and our proposed ResZero, with GT and Majority-Voting as references. As shown in Table 3, among all proof-gated variants ResZero consistently achieves the highest *Average*. On average, it outperforms Zero Reward by **+1.3** pts, MV Reward by **+6.1** pts, and essentially

Table 3: Ablation results for the proposed ResZero reward ($\delta = 0$) on reasoning benchmark.

| Methods | Mathematics | | | | | Code | | Instruction | Multi-Task | Average |
|---|---|---|---|---|---|---|---|---|---|---|
| | AIME24 | AIME25 | MATH-500 | GSM8K | AMC | LiveCode | CRUX | IFEval | MMLU-Pro | |
| *Qwen3-1.7B-Base* | | | | | | | | | | |
| GT-Reward | $6.88_{\pm2.1}$ | $5.21_{\pm2.0}$ | $68.35_{\pm2.3}$ | $82.01_{\pm1.1}$ | $30.87_{\pm3.3}$ | $14.84_{\pm0.5}$ | $33.73_{\pm0.6}$ | $38.70_{\pm1.6}$ | $39.43_{\pm0.7}$ | 35.56 |
| Majority-Voting | $2.08_{\pm1.3}$ | $2.08_{\pm1.3}$ | $59.6_{\pm0.9}$ | $81.08_{\pm0.8}$ | $29.82_{\pm1.7}$ | $14.29_{\pm0.5}$ | $32.00_{\pm2.3}$ | $37.16_{\pm1.6}$ | $35.68_{\pm0.7}$ | 32.64 |
| Proof-Gate + Zero Reward | $3.54_{\pm1.6}$ | $2.92_{\pm1.4}$ | $59.85_{\pm1.2}$ | $83.21_{\pm0.5}$ | $30.87_{\pm2.4}$ | $15.00_{\pm0.8}$ | $34.10_{\pm1.4}$ | $37.40_{\pm1.6}$ | $34.43_{\pm0.7}$ | 33.48 |
| Proof-Gate + Random Reward | $4.58_{\pm1.4}$ | $2.50_{\pm1.2}$ | $59.75_{\pm2.0}$ | $77.58_{\pm0.8}$ | $26.96_{\pm1.3}$ | $9.14_{\pm1.3}$ | $18.48_{\pm1.6}$ | $37.19_{\pm1.6}$ | $35.12_{\pm0.7}$ | 30.14 |
| Proof-Gate + MV Reward | $0.00_{\pm0.0}$ | $0.00_{\pm0.0}$ | $41.2_{\pm1.0}$ | $82.47_{\pm0.4}$ | $9.64_{\pm0.8}$ | $14.48_{\pm1.1}$ | $34.40_{\pm2.1}$ | $36.78_{\pm1.6}$ | $35.33_{\pm0.7}$ | 28.26 |
| **JURY-RL (Proof-Gate + ResZero)** | $7.71_{\pm1.8}$ | $4.58_{\pm2.0}$ | $66.4_{\pm2.0}$ | $82.58_{\pm0.3}$ | $34.19_{\pm3.1}$ | $14.54_{\pm0.4}$ | $33.98_{\pm2.5}$ | $36.55_{\pm1.6}$ | $38.11_{\pm0.7}$ | 35.40 |
| *Llama-3.2-3B-Instruct* | | | | | | | | | | |
| GT-Reward | $10.62_{\pm2.1}$ | $0.21_{\pm0.4}$ | $47.5_{\pm1.4}$ | $78.60_{\pm0.8}$ | $22.89_{\pm1.9}$ | $7.05_{\pm0.6}$ | $32.48_{\pm0.8}$ | $50.16_{\pm1.8}$ | $34.26_{\pm0.7}$ | 31.53 |
| Majority-Voting | $8.33_{\pm1.3}$ | $0.00_{\pm0.0}$ | $48.4_{\pm3.0}$ | $78.64_{\pm1.4}$ | $21.69_{\pm2.8}$ | $8.11_{\pm0.5}$ | $30.83_{\pm1.4}$ | $48.36_{\pm1.8}$ | $33.94_{\pm0.7}$ | 30.92 |
| Proof-Gate + Zero Reward | $8.33_{\pm1.6}$ | $0.42_{\pm0.6}$ | $47.8_{\pm2.2}$ | $78.54_{\pm0.8}$ | $20.48_{\pm3.2}$ | $4.36_{\pm0.5}$ | $30.70_{\pm1.1}$ | $49.98_{\pm1.7}$ | $35.06_{\pm0.7}$ | 30.63 |
| Proof-Gate + Random Reward | $2.92_{\pm1.4}$ | $1.25_{\pm1.1}$ | $41.85_{\pm2.2}$ | $72.71_{\pm2.9}$ | $17.32_{\pm2.1}$ | $3.03_{\pm0.8}$ | $26.88_{\pm1.1}$ | $52.30_{\pm1.8}$ | $31.90_{\pm0.7}$ | 27.80 |
| Proof-Gate + MV Reward | $7.29_{\pm1.4}$ | $0.00_{\pm0.0}$ | $47.25_{\pm3.3}$ | $78.28_{\pm0.6}$ | $21.99_{\pm2.6}$ | $7.07_{\pm1.0}$ | $30.87_{\pm0.7}$ | $48.49_{\pm1.7}$ | $33.68_{\pm0.7}$ | 30.55 |
| **JURY-RL (Proof-Gate + ResZero)** | $9.17_{\pm2.0}$ | $1.25_{\pm0.9}$ | $48.8_{\pm2.6}$ | $78.96_{\pm0.5}$ | $26.05_{\pm1.6}$ | $6.16_{\pm0.6}$ | $31.60_{\pm1.2}$ | $50.09_{\pm1.7}$ | $34.54_{\pm0.7}$ | 31.85 |
| *Qwen2.5-7B* | | | | | | | | | | |
| GT-Reward | $18.33_{\pm2.3}$ | $11.25_{\pm1.7}$ | $75.1_{\pm1.4}$ | $90.83_{\pm0.5}$ | $46.54_{\pm2.5}$ | $12.78_{\pm1.6}$ | $53.67_{\pm1.0}$ | $41.50_{\pm1.7}$ | $45.09_{\pm0.7}$ | 43.90 |
| Majority-Voting | $11.25_{\pm1.2}$ | $11.25_{\pm1.7}$ | $71.0_{\pm0.7}$ | $90.52_{\pm0.7}$ | $38.70_{\pm1.3}$ | $18.37_{\pm0.9}$ | $52.20_{\pm1.1}$ | $42.72_{\pm1.7}$ | $43.83_{\pm0.7}$ | 41.42 |
| Proof-Gate + Zero Reward | $11.67_{\pm1.7}$ | $10.21_{\pm2.9}$ | $75.15_{\pm1.8}$ | $90.13_{\pm0.5}$ | $41.11_{\pm3.0}$ | $18.10_{\pm1.2}$ | $52.02_{\pm1.8}$ | $43.67_{\pm1.7}$ | $46.97_{\pm0.8}$ | 43.23 |
| Proof-Gate + Random Reward | $8.75_{\pm1.9}$ | $2.50_{\pm1.2}$ | $60.7_{\pm2.0}$ | $82.66_{\pm1.7}$ | $30.72_{\pm4.5}$ | $21.97_{\pm1.2}$ | $43.50_{\pm1.1}$ | $38.56_{\pm1.7}$ | $43.90_{\pm0.7}$ | 37.03 |
| Proof-Gate + MV Reward | $0.00_{\pm0.0}$ | $0.00_{\pm0.0}$ | $51.4_{\pm1.1}$ | $90.54_{\pm1.0}$ | $14.76_{\pm1.0}$ | $17.29_{\pm1.0}$ | $54.83_{\pm1.5}$ | $43.64_{\pm1.7}$ | $35.01_{\pm0.7}$ | 34.16 |
| **JURY-RL (Proof-Gate + ResZero)** | $13.75_{\pm1.8}$ | $7.50_{\pm1.6}$ | $75.65_{\pm0.7}$ | $90.35_{\pm0.9}$ | $47.29_{\pm3.2}$ | $14.69_{\pm1.1}$ | $54.37_{\pm0.4}$ | $41.51_{\pm1.7}$ | $50.00_{\pm0.7}$ | 43.90 |

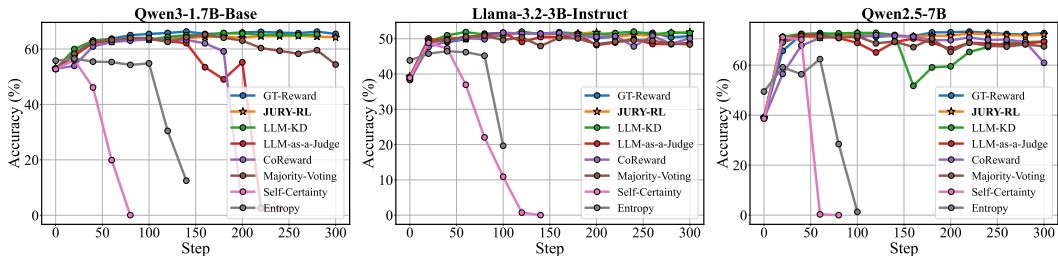

Figure 3: Accuracy on MATH5000 Validation set over training steps.

matches GT within **+0.1** pts, while surpassing the Random Reward baseline by **+5.4** pts. Notably, *Proof-Gate + MV Reward* performs worse than the plain Majority-Voting baseline. This highlights that rewarding an unverified majority can reinforce spurious consensus even after verification fails, accelerating mode collapse. A simple Zero Reward achieves suboptimal performance by taking a safe but inefficient path: it avoids reinforcing errors, but at the cost of stalling the learning process for inconclusive samples. Meanwhile, the naive Random Reward baseline, which injects unstructured reward noise when verification fails, underperforms Zero Reward by **4.1** pts on average, indicating that random reinforcement alone cannot provide a useful learning signal. ResZero provides a robust solution by navigating this trade-off between dangerous reinforcement and inefficient stagnation. Per-benchmark pass@k breakdowns can be found in Table 9, Appendix G.1, where ResZero significantly outperforms other variants and even surpasses the GT-Reward baseline across all models.

**JURY-RL Achieves Stable Training and Avoids Collapse.** To evaluate training stability, we tracked the validation accuracy of JURY-RL against label-free baselines on MATH5000 validation set throughout the training process. As illustrated in Figure 3, The Entropy and Self-Certainty show collapse after an initial performance gain, as the model begins to reinforce spurious consensus. The LLM-as-a-Judge/LLM-KD/Majority-Voting exhibit noisy and suboptimal convergence. In contrast, JURY-RL demonstrates stable, monotonic improvement, confirming that its proof-gated reward mechanism effectively prevents the mode collapse common in self-supervised methods.

**Analysis of Verifier Signal Quality.** Is Lean a better judge than LLM? While a formal verifier like Lean theoretically offers near-zero false positives, our practical pipeline involves upstream processes like auto-formalization and consistency checks, which can introduce errors. It is therefore crucial to compare the signal quality of our verifier against an LLM-as-a-Judge. As shown in Table 4, our Lean verifier provides a superior reward signal compared to the LLM-as-a-Judge. It achieves substantially higher **precision** (84.5% vs. 75.9%) at the cost of moderately lower recall (88.0% vs. 96.1%). This trade-off is paramount: high precision drastically reduces false positives,

preventing reward hacking and tightly aligning the training objective with verifiable correctness. Conversely, the LLM-judge's noisy signal, stemming from low precision, risks reinforcing errors despite its higher recall. The Lean verifier's higher F1-score (86.2%) confirms its better overall balance, validating our "Proofs Dispose" principle of prioritizing signal fidelity for stable learning. While our verifier is not perfect, its

Table 4: Verifier signal quality on training set. All metrics reported in percent (%).

| Verifier | Prec. | Rec. | F1 |
|---|---|---|---|
| LLM-as-a-Judge | 75.9 | 96.1 | 84.8 |
| Lean Verifier (Ours) | **84.5** | 88.0 | **86.2** |

imperfections stem from upstream components rather than the prover's core logic. We provide a deeper analysis of these nuances and their effect on training dynamics in Appendix G.3.

**Impact of $c$.** We analyze the impact of the hyperparameter $c$ in Eq. 5, which controls the penalty strength on the unverified majority proposal in our ResZero reward. As illustrated in Figure 4, $c$ is critical in navigating the trade-off between preventing mode collapse and maximizing task performance. The right panel clearly demonstrates that a non-zero $c$ is essential for maintaining solution diversity. When $c = 0$, the framework effectively degenerates to a zero-reward fallback, leading to a sharp decline in the average number of unique answers as training progresses—a classic symptom of the model converging to a spurious consensus. In contrast, any positive $c$ value successfully sustains a high level of diversity. However, the left and center panels reveal a subtle trade-off: an overly aggressive penalty (e.g., $c = 0.1$) can slightly suppress the final reward and accuracy. This suggests that while the penalty is crucial for exploration, an excessive value may overly restrict the policy from exploiting a potentially correct, high-consensus answer. This ablation validates that a moderately tuned $c$ (e.g., $c = 0.01$ in our experiments) strikes an optimal balance, ensuring robust training stability and solution diversity without compromising convergence on the primary task objectives.

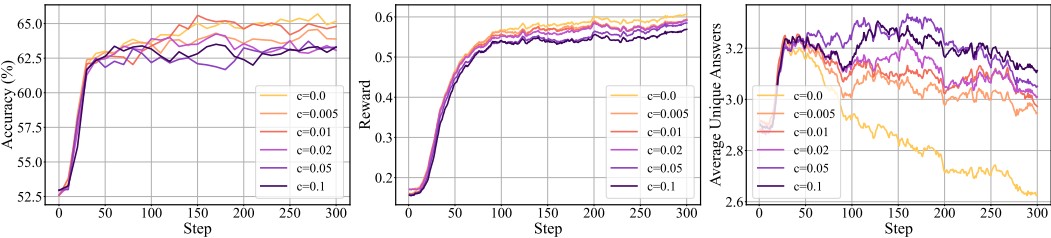

Figure 4: Training dynamics under different values of the hyperparameter $c$.

## 6 CONCLUSION

We introduced **JURY-RL**, a label-free RLVR framework that decouples *proposal* from *disposal*: majority voting across rollouts merely proposes a candidate answer, while a formal Lean verifier disposes the final reward. If the proposal is verified, only the supporting trajectories are rewarded; when verification is inconclusive, **ResZero** discards the unverifiable majority and assigns a zero-mean, variance-preserving residual reward that keeps optimization well-conditioned. This design jointly achieves three goals: scalability without human labels, truth alignment via verifiable correctness, and optimization stability in the absence of proof. Across mathematical reasoning, code generation, and multi-task evaluations, JURY-RL trains more stably than label-free baselines and achieves performance comparable to supervised training with ground-truth rewards, while demonstrating more promising diversity and pass@k results. Our work demonstrates that grounding RL in sparse but formally verified signals is a promising strategy for building robust and generalizable reasoning models without human labels.

## ETHICS STATEMENT

This work studies label-free reinforcement learning with verifiable rewards (RLVR) for mathematical and programmatic reasoning. It does not involve human subjects, crowd workers, user studies, or the collection of personally identifiable information. All datasets used (e.g., MATH, GSM8K, AMC/AIME, LiveCodeBench, CRUX, MMLU-Pro, IFEval) are publicly available and used under their

respective licenses; we redistribute nothing and provide only references and scripts to download from the original sources.

Potential risks include (i) *misuse*: stronger automated reasoning could be used to complete graded assignments or to generate convincing but incorrect solutions; (ii) *bias/coverage*: public benchmarks may contain stylistic biases or limited topical coverage; (iii) *safety*: LLM judges can transfer prompt or format biases into training signals. Our design explicitly mitigates these issues by proof-gating rewards with a formal verifier (Lean) to reduce false positives and by using the Residual-Zero (ResZero) fallback to avoid reinforcing unverifiable consensus. We report precisely which datasets, prompts, and evaluation harnesses are used, and we provide ablations and diagnostics to surface failure modes (e.g., collapse under majority-voting). We release only code, configuration files, and download scripts; no private or proprietary data are included.

## REPRODUCIBILITY STATEMENT

We are committed to ensuring the full reproducibility of our research. All necessary components, including code, models, datasets, and experimental settings, are detailed below.

**Theory.** Full proofs for our theoretical analysis (including the zero-mean property of RESZERO) are provided in Appendix A.

**Code.** The implementation of JURY-RL and all baseline methods is based on the publicly available VeRL framework (Sheng et al., 2025). The complete source code for our experiments, including scripts for training and evaluation, will be made publicly available upon publication. All evaluation frameworks used are standard and open-source, with specific details and links provided in Appendix F.

**Models.** The backbone models used in our experiments are all publicly available open-source models from the Qwen and Llama3 series. Specifically, we used **Qwen3-1.7B-Base**, **Llama-3.2-3B-Instruct**, and **Qwen2.5-7B**. These models can be accessed through official repositories such as Hugging Face. The judge model used in our LLM-as-a-Judge baseline is `qwen-2.5-72b-instruct`, which is also publicly accessible. We will release our trained model upon acceptance.

**Datasets.** All datasets used for training and evaluation are standard, publicly available benchmarks. We train our models on a 7,500-problem subset of the official **MATH** training split (Hendrycks et al., 2021). We evaluate on the following benchmarks: **AIME24/25**, **MATH500**, **GSM8K**, **AMC**, **LiveCodeBench**, **CRUX**, **IFEval**, and **MMLU-Pro**. References and links for each are provided in Section 5.1.

**Experimental Setup and Hyperparameters.** All crucial hyperparameters for training, optimization, and generation are provided in Table 7 in Appendix E. The prompt formats used for all models are their officially released chat-based formats to ensure faithful reproduction. Details of the baseline implementations are described in Appendix D. The design of our Lean verifier is detailed in Appendix C.

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

# APPENDIX

## A  THEORETICAL ANALYSIS

This section gives a minimal, checkable account of why Jury-RL stabilizes policy optimization.

- **§A.1 — Why naive majority voting is brittle.** We show that, under GRPO, majority-vote rewards conflate agreement with correctness: as the majority share increases, supporters' advantages go to zero while dissenters' penalties blow up, driving entropy collapse.
- **§A.2 — ResZero is zero-mean with non-degenerate variance.** We prove the group reward strictly sums to zero with the choice $\gamma = c\alpha^2$, and that the group variance is non-zero whenever residual answers are not all identical, so gradients remain informative when verification fails (Eq. 5).
- **§A.3 — Fallback comparison (MV / Zero / ResZero).** We derive group-normalized advantages for each fallback and show that only ResZero yields a corrective (negative for unverifiable majorities) yet exploratory (variance preserved on residuals) update, aligning with the stability observed in §5.3.

### A.1  WHY NAIVE MAJORITY VOTING IS BRITTLE.

We retain the previously derived analysis showing that majority voting (MV) conflates agreement with correctness and induces entropy collapse under GRPO as consensus strengthens. Using MV as a label-free reward conflates single-view agreement with correctness and induces entropy collapse. For a question $x$, sample $G$ rollouts $y_i \sim \pi_\theta(\cdot \mid x)$ and parse answers $a_i = \text{ans}(y_i)$. Let $\hat{a} = \arg\max_a \sum_{j=1}^G \mathbb{I}[a_j = a]$ with vote share $\hat{v} = \frac{1}{G}\sum_{j=1}^G \mathbb{I}[a_j = \hat{a}]$. MV assigns binary rewards $r_i^{\text{MV}} = \mathbb{I}[a_i = \hat{a}]$. Under GRPO, group-normalized advantages satisfy $\overline{r} = \hat{v}$, $\text{std} = \sqrt{\hat{v}(1-\hat{v})}$, hence

$$\hat{A}_i = \frac{r_i^{\text{MV}} - \overline{r}}{\text{std}} = \begin{cases} \sqrt{\frac{1-\hat{v}}{\hat{v}}}, & a_i = \hat{a}, \\ -\sqrt{\frac{\hat{v}}{1-\hat{v}}}, & a_i \neq \hat{a}. \end{cases}$$

As a spurious consensus strengthens ($\hat{v} \to 1$), supporters receive vanishing signal ($\hat{A}_i^+ \to 0$) while dissenters incur diverging penalties ($\hat{A}_i^- \to -\infty$), suppressing exploration and shrinking the token-level entropy toward a single mode; ratio clipping preserves the sign of $\hat{A}_i$, so the collapse persists. Because $r_i^{\text{MV}}$ ignores ground truth, MV is also vulnerable to formatting hacks: repeated insertion of a frequent symbol in the "answer box" can maximize agreement without correctness.

### A.2  PROOF OF ZERO-MEAN PROPERTY FOR RESZERO REWARD

**Proposition (Zero-mean property of ResZero).**   Consider any group of $G \geq 1$ trajectories, with majority set $M$, residual set $R$, and majority share $\alpha = |M|/G$. Let the ResZero reward $r_i^{\text{ResZero}}$ be defined as in Equation (5) with $\gamma = c\alpha^2$. Then, for all possible vote patterns (including ties and degenerate cases),

$$\sum_{i=1}^G r_i^{\text{ResZero}} = 0.$$

This property is critical for ensuring stable optimization for different RL methods.

**Edge cases.** *(i) Tie-breaking)* When multiple answers tie for the most frequent, we select a single $\hat{a}$ by a fixed rule, so $M = \{i : a_i = \hat{a}\}$ and $R = \{i : a_i \neq \hat{a}\}$ are uniquely defined. *(ii) No residuals)* If all $G$ trajectories coincide, then $|R| = 0$, $|M| = G$, and the residual sum is zero. The total reward becomes $\sum_{i \in M} r_i^{\text{ResZero}} = |M|(-c\alpha + \gamma)$. With $\alpha = |M|/G = 1$ and $\gamma = c\alpha^2 = c$, this equals $G(-c + c) = 0$.

We now present the proof for the general case where $|R| > 0$. The total sum of rewards is decomposed as:

$$\sum_{i=1}^{G} r_i^{\text{ResZero}} = \sum_{i \in M} r_i^{\text{ResZero}} + \sum_{i \in R} r_i^{\text{ResZero}}$$

**Step 1: Calculate the sum of rewards for the residual group (R).** For any trajectory $i \in R$, the reward is $r_i^{\text{ResZero}} = \alpha \cdot (z_i - \bar{u}) + \gamma$. Summing over all members of the residual group:

$$\sum_{i \in R} r_i^{\text{ResZero}} = \sum_{i \in R} [\alpha(z_i - \bar{u}) + \gamma]$$

$$= \alpha \sum_{i \in R} (z_i - \bar{u}) + \sum_{i \in R} \gamma$$

By the definition of a mean, the sum of deviations from the mean is zero, i.e., $\sum_{i \in R}(z_i - \bar{u}) = 0$. This simplifies the total reward for the residual group to:

$$\sum_{i \in R} r_i^{\text{ResZero}} = \alpha \cdot 0 + |R| \cdot \gamma = |R|\gamma$$

**Step 2: Calculate the sum of rewards for the majority group (M).** For any trajectory $i \in M$, the reward is $r_i^{\text{ResZero}} = -c\alpha + \gamma$. Since the reward is identical for all members of the majority group, the sum is:

$$\sum_{i \in M} r_i^{\text{ResZero}} = |M| \cdot (-c\alpha + \gamma)$$

**Step 3: Combine the sums from both groups.** We add the sums from Step 1 and Step 2 to find the total sum:

$$\sum_{i=1}^{G} r_i^{\text{ResZero}} = |M|(-c\alpha + \gamma) + |R|\gamma$$

$$= -c\alpha|M| + (|M| + |R|)\gamma$$

Since $|M| + |R| = G$, the equation becomes:

$$\sum_{i=1}^{G} r_i^{\text{ResZero}} = -c\alpha|M| + G\gamma$$

**Step 4: Derivation of the Global Re-centering Term $\gamma$.** To enforce the zero-mean property, we **design** the global re-centering term $\gamma$ such that the total sum from Step 3 is identically zero.

$$-c\alpha|M| + G\gamma = 0$$

$$\gamma = \frac{c\alpha|M|}{G}$$

By substituting the definition of the majority share, $\alpha = |M|/G$, we arrive at the required form for $\gamma$:

$$\gamma = c\alpha \left( \frac{|M|}{G} \right) = c\alpha^2$$

This derivation shows that setting $\gamma = c\alpha^2$ is the precise design choice required to make the ResZero reward strictly zero-centered. This term acts as a global offset that exactly balances the penalties applied to the majority group and the rewards distributed among the residual group.

### A.3 FALLBACK REWARDS FOR INCONCLUSIVE VERIFICATION

This section provides a theoretical analysis of different fallback reward mechanisms within the GRPO framework for the scenario where formal verification of the majority-voted answer is inconclusive ($\delta = 0$). The analysis focuses on the dynamics of the group-normalized advantage, $\hat{A}_i$, which serves as the learning signal for the policy update. The advantage is defined as:

$$\hat{A}_i = \frac{r_i - \bar{r}}{\text{std}(\{r_j\}_{j=1}^G) + \varepsilon}, \quad \text{where} \quad \bar{r} = \frac{1}{G} \sum_{j=1}^G r_j.$$

We analyze three cases: rewarding the majority vote (MV), assigning a zero reward, and using our proposed ResZero reward.

#### CASE 1: MAJORITY VOTING (MV) REWARD

As established in Appendix A.1, if we naively reward the majority consensus when verification fails, the reward is $r_i^{\text{MV}} = \mathbb{I}[a_i = \hat{a}]$. Let $\hat{v}$ be the vote share of the majority answer $\hat{a}$. The key statistics are $\bar{r} = \hat{v}$ and $\text{std}(\{r_j\}) = \sqrt{\hat{v}(1-\hat{v})}$. This yields the following advantage:

$$\hat{A}_i = \begin{cases} \sqrt{\frac{1-\hat{v}}{\hat{v}}}, & \text{if } a_i = \hat{a} \text{ (Supporter)} \\ -\sqrt{\frac{\hat{v}}{1-\hat{v}}}, & \text{if } a_i \neq \hat{a} \text{ (Dissenter)} \end{cases}$$

**Theoretical Implication.** As a spurious consensus strengthens ($\hat{v} \to 1$), the advantage for supporters vanishes ($\hat{A}_i^+ \to 0$) while the penalty for dissenters diverges ($\hat{A}_i^- \to -\infty$). This dynamic punishes any exploration and provides no positive signal for adhering to the consensus, leading to **entropy collapse** and policy degradation.

#### CASE 2: ZERO REWARD

A seemingly safe alternative is to assign a zero reward to all rollouts when verification is inconclusive. In this case, $r_i^{\text{Zero}} = 0$ for all $i \in \{1, \dots, G\}$.

The resulting statistics are trivial:

- **Mean Reward:** $\bar{r} = \frac{1}{G} \sum_{i=1}^G 0 = 0$.

- **Standard Deviation:** $\text{std}(\{r_j\}) = \sqrt{\frac{1}{G} \sum_{i=1}^G (0-0)^2} = 0$.

Substituting these into the advantage formula gives:

$$\hat{A}_i = \frac{0-0}{0+\varepsilon} = 0$$

**Theoretical Implication.** The advantage signal is nullified for all rollouts in the group. This leads to a **vanishing gradient** for the policy update, effectively stalling the learning process for that entire batch. While it avoids the destructive collapse of MV, it does so at the cost of learning efficiency, rendering the update step ineffective.

#### CASE 3: RESZERO REWARD

The ResZero reward is designed specifically to address the shortcomings of the above methods. It has two crucial properties by construction:

- **Zero-Mean Property:** The total reward across the group sums to zero, i.e., $\sum_{i=1}^G r_i^{\text{ResZero}} = 0$. This immediately implies the mean reward is $\bar{r} = 0$.
- **Non-Zero Variance:** By assigning a negative reward to the majority group and a structured, zero-mean reward to the residual group, $r_i^{\text{ResZero}}$ is non-zero for most rollouts (unless all residual answers are identical). Therefore, $\text{std}(\{r_j\}) > 0$ as long as there is diversity among residual answers.

The advantage function thus becomes:

$$\hat{A}_i = \frac{r_i^{\text{ResZero}} - 0}{\text{std}(\{r_j\}) + \varepsilon} = \frac{r_i^{\text{ResZero}}}{\text{std}(\{r_j\}) + \varepsilon}$$

**Theoretical Implication.** This formulation establishes a stable, self-regulating update dynamic that avoids the extremes of vanishing gradients (Case 2) or diverging penalties (Case 1). Intuitively, ResZero operates as a **damped correction mechanism**:

- **Adaptive Penalization:** If the model is *uncertain* (small majority share $\alpha$), the corrective signal is mild. However, if the model is *confidently wrong* (large $\alpha$ but unverified), both the majority penalty and residual amplification scale up. This specifically targets spurious consensus without destabilizing early training.

- **Stability against Oscillations:** Unlike a binary flip, ResZero redistributes probability mass toward residuals proportional to their internal support $z_i$ (as defined in §A.2). This creates a zero-mean, variance-preserving gradient that gently pushes the policy toward a more balanced mixture rather than inducing unbounded oscillations.

This mechanism explains the stability observed in our experiments (§5.3): ResZero provides a directional signal to undo unverified confidence while GRPO ensures the updates remain bounded.

In summary, our theoretical analysis reveals that each fallback strategy results in a fundamentally different learning dynamic:

- **Majority Voting (MV)** leads to a **destructive update**. As a spurious consensus strengthens, it creates a diverging penalty for dissent while the positive signal for supporters vanishes. This dynamic ultimately causes **entropy collapse**, suppressing exploration and degrading the policy.

- **Zero Reward** results in an **ineffective update**. By nullifying the reward for all rollouts, the advantage signal becomes zero for the entire group. This causes a **vanishing gradient** that stalls the learning process for that step, wasting computational resources.

- **ResZero Reward** provides a **constructive update**. It maintains a stable, non-zero learning signal that is both **corrective**, by penalizing the unverified majority consensus, and **exploratory**, by rewarding promising alternatives among the residual answers.

# B  IMPLEMENTATION DETAILS

## B.1  THE JURY-RL ALGORITHM

The full algorithm is summarized in Algorithm 1.

---

**Algorithm 1** JURY-RL (one grouped update for a prompt $x$)

---

1: Sample $G$ rollouts $y_i \sim \pi_{\text{old}}(\cdot \mid x)$; parse $a_i = \text{ans}(y_i)$.
2: Compute vote shares $v(a)$ and majority $\hat{a} \in \arg\max_a v(a)$.
3: Query verifier once: $\delta = \text{verify}(x, \hat{a})$.
4: **if** $\delta=1$ **then**
5:     Set $r_i = \mathbb{I}[a_i = \hat{a}]$.
6: **else**
7:     Form $M, R$; compute $u^{(-i)}$ and $\bar{u}$; set $r_i$ via Eq. (5).
8: **end if**
9: Compute group-normalized advantages $\hat{A}_i$ from $\{r_i\}_{i=1}^G$; broadcast token-wise.
10: Update $\pi_\theta$ with GRPO (clipped ratios, KL to reference).

---

### B.2 An Intuitive Example of the ResZero Reward

Consider a scenario with $G = 8$ rollouts for a given problem. A majority vote reveals that $|M| = 4$ rollouts support a proposal $\hat{a}$, so the majority share is $\alpha = |M|/G = 4/8 = 0.5$. The remaining $|R| = 4$ residual rollouts are split: three support answer $b$, and one supports answer $c$. If the verifier is inconclusive ($\delta = 0$), the ResZero reward is activated.

First, we analyze the residual group $R$. For a trajectory $y_i$ with answer $a_i = b$, its leave-one-out support within the residual group is $z_i = u^{(-i)}(b) = \frac{2}{|R|-1} = 2/3$. For the singleton answer $a_i = c$, the support is $z_i = u^{(-i)}(c) = 0/3 = 0$. The average support across the residual group is therefore $\bar{u} = \frac{1}{|R|}(3 \cdot \frac{2}{3} + 1 \cdot 0) = 0.5$.

Let the penalty hyperparameter be $c = 0.1$. The global re-centering term is $\gamma = c\alpha^2 = 0.1 \cdot (0.5)^2 = 0.025$. The final reward for each trajectory is assigned according to Eq. (5):

$$
r_i^{\text{ResZero}} = \begin{cases} -c\alpha + \gamma = -0.1(0.5) + 0.025 = -0.025, & \text{if } i \in M \text{ (proposal } \hat{a}) \\ \alpha(z_i - \bar{u}) + \gamma = 0.5(\frac{2}{3} - 0.5) + 0.025 = \frac{1}{12} + 0.025 \approx 0.0667, & \text{if } i \in R, a_i = b \\ \alpha(z_i - \bar{u}) + \gamma = 0.5(0 - 0.5) + 0.025 = -0.225, & \text{if } i \in R, a_i = c \end{cases}
$$

By design, the total reward sums exactly to zero: $4(-0.025) + 3(\frac{1}{12} + 0.025) + 1(-0.225) = -0.1 + 0.25 + 0.075 - 0.225 = 0$. This demonstrates how ResZero penalizes the unverifiable majority while redistributing a zero-mean signal among diverse residual answers. This maintains a useful, variance-driven learning gradient for GRPO without reinforcing a potentially spurious consensus.

## C Lean Verifier Details

### C.1 Lean Verifier

Lean (de Moura & Ullrich, 2021) has emerged as a transformative framework in the formal verification of mathematical proofs, grounded in a rigorous type-theoretic foundation that guarantees unprecedented levels of logical soundness and mechanized reliability. Its intrinsic dependency on computer-assisted compilation environments not only ensures formal correctness but also serves as a high-fidelity feedback mechanism for refining and validating mathematical reasoning within LLMs.

In response to this potential, we have designed and implemented a comprehensive mathematical verification system centered on Lean. This system bridges the gap between informal natural language mathematics and machine-verifiable formalism by automatically translating problem statements and proposed solutions into syntactically and semantically well-formed Lean expressions, followed by formal proof verification via Lean's trusted kernel.

The architecture of this verification system ( illustrated in Figure 5 ) is a cascaded, modular pipeline comprising three specialized components:

- **Autoformalizer** Translates natural language mathematical content (question—answer pairs in our setting) into precise, executable Lean formal specifications, preserving both syntactic structure and semantic intent.

- **Consistency-Checker** Performs bidirectional semantic alignment between the original input of natural language and its formalized Lean counterpart, ensuring fidelity of meaning and detecting potential misinterpretations or translation artifacts.

- **Prover** Synthesizes formal proof scripts in Lean and submits them to the theorem prover for mechanized validation, thus certifying the logical correctness of the solution under the foundational logic of the system.

This framework not only advances the automation of mathematical reasoning verification, but also establishes a scalable, feedback-driven paradigm for integrating formal methods into natural language-based mathematical systems.

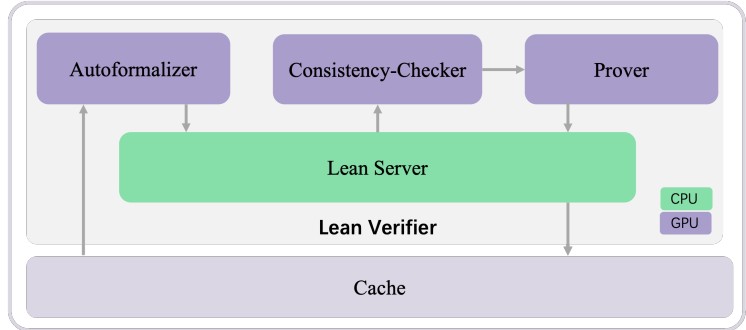

Figure 5: The pipeline of our Lean Verifier: a collaborative system that combines autoformalization, formal verification, and consistency checking to validate mathematical reasoning with high reliability.

## C.2 SETTING

To ensure reliable and precise reward signals, our Lean Verifier employs a three-stage inference pipeline composed of the following core components:

- **Autoformalizer**: Utilizes the Goedel-Formalizer-V2-32B model (Lin et al., 2025) to translate informal mathematical statements into formal Lean specifications.

- **Consistency-Checker**: Implemented using QwQ-32B (QwenTeam, 2025), this component validates the outputs of the Autoformalizer.

- **Prover**: Employs the Goedel-Prover-V2-32B model (Lin et al., 2025) to synthesize formal proofs for the specifications that have been validated in the previous stage.

The pipeline operates as follows: first, the Autoformalizer generates up to 8 candidate formalizations. The Consistency-Checker then evaluates effective candidates and selects one checked candidate. Finally, the Prover conducts up to 16 independent sampling trials to find a valid proof.

All formal verification outputs were generated under the same hyperparameter configuration below:

- **Temperature**: 0.7
- **Maximum Tokens**: 32,768

We make the configuration choice to balance exploration, fidelity, and resource efficiency.

## C.3 COMPONENT-WISE ACCURACY ANALYSIS

To better understand the reliability of our reward signal, we evaluated the independent performance of the Autoformalizer and Consistency Checker on the MATH500 dataset.

**Autoformalizer Reliability.** We evaluated the autoformalization success rate. The Autoformalizer successfully generated syntactically valid and type-checkable Lean 4 statements for **477 out of 500 problems**, yielding a success rate of **95.4%**. This high coverage ensures that the vast majority of mathematical problems can be converted into a verifiable format.

**Consistency Checker Accuracy.** The Consistency Checker is responsible for filtering out formalizations that are syntactically valid but semantically inconsistent with the original problem. As shown in Table 5, the model achieves an F1-score of **91.1%** and a Recall of **94.9%**, demonstrating its effectiveness in maintaining semantic fidelity without aggressively discarding valid candidates.

Combined with the formal soundness of the Lean Prover (which has a false positive rate of 0%), these components ensure that JURY-RL receives a reward signal that is both high-coverage and high-precision.

Table 5: Performance metrics of our Consistency Checker.

| Metric | Accuracy | Precision | Recall | F1-score |
|---|---|---|---|---|
| Consistency Checker | 85.3% | 87.7% | 94.9% | 91.1% |

## C.4 PERFORMANCE AND COST ANALYSIS

A crucial factor for the practical implementation of JURY-RL is the computational overhead introduced by the formal Lean Verifier, especially when compared to baselines like an LLM-as-a-Judge. To provide a clear and quantitative assessment of this cost, we conducted a wall-clock time analysis using the Qwen-2.5-7B model. The evaluation was performed under the synchronous RL training configuration specified in our main paper.

**Initial (Cold-Start) Overhead.** To accurately assess the computational cost, we first establish the baseline training time. The oracle setting with Ground-Truth (GT) rewards, which involves no external verification, requires approximately **100 seconds** per step. This duration represents the core training workload.

We then measured the additional verification overhead for the other methods under a worst-case scenario, defined as processing a batch composed **entirely of unseen question-answer (QA) pairs**. The specific overheads are as follows:

- **JURY-RL (Lean Verifier):** Under the shared setting where each training step processes 128 questions and the verifier is called once per question on the majority-vote answer, the Lean-based verification adds an overhead of approximately **200 seconds per step**.
- **LLM-as-a-Judge (Qwen-2.5-72B):** Using the judge to score only the majority-vote answer for each of the 128 questions adds an overhead of approximately **80 seconds per step**.

This means that in the initial training stages, the total step time for JURY-RL is approximately 300 seconds (100s for training + 200s for verification), compared to 180 seconds for the LLM-as-a-Judge. It is important to note that the efficiency of JURY-RL is significantly bolstered by an **early-stopping mechanism**. Although our framework attempts to autoformalize and prove the statement up to $k$ times (in the $pass@k$ setting), the process terminates immediately upon the first success. Since we verify the consensus answer derived from majority voting—which has a high probability of correctness—the prover often succeeds in the very first few attempts, avoiding the computational cost of executing all $k$ trials.

**Cost Amortization and Convergence.** The initial cold-start overhead is not representative of the average cost over the entire training process. Our framework uses a caching mechanism for all verification results (shown in Fig.5). As training proceeds and the policy begins to converge, the diversity of generated answers for any given problem stabilizes. Consequently, an increasing fraction of QA pairs in subsequent batches will have been previously encountered and verified. These cached results can be retrieved almost instantly, bypassing the expensive formal verification process. In the steady-state phase of training, this caching effect becomes dominant. The per-step time cost for JURY-RL progressively converges toward that of the GT baseline, as most verification lookups are resolved via the cache. Therefore, while the initial cost of the Lean Verifier is higher, this cost is effectively amortized throughout the training run. We contend that this represents an acceptable and practical trade-off for the substantial gains in reward fidelity, training stability, and final model performance documented in our main results.

**Comparison with LLM-as-a-Judge.** Beyond the runtime overhead, another critical aspect of the verifier is the trade-off between its performance and consumption, which directly impacts both the quality of the reward signal and the operational cost. To analyze this, we evaluated different prover configurations on the MATH500 dataset and compared them with an LLM-as-a-judge baseline.

Table 6 summarizes the performance, model size, and runtime characteristics of these configurations. Among the Lean-based verifiers, the Prover@64 configuration achieves the highest accuracy with

the largest end-to-end evaluation time on MATH500 (655.80 s). In contrast, the Prover@16 configuration, while exhibiting only a slight and acceptable decrease in performance to 87.6% ACC, reduces the total runtime to 324.31 s. Given this highly favorable trade-off between a marginal performance drop and substantial cost savings, we adopt Prover@16 as the default Lean-based configuration for our main experiments. For comparison, Table 6 also reports an LLM-as-a-judge variant based on Qwen2.5-72B-instruct with four-way majority voting (mv@4). This judge uses a substantially larger model (72B versus 32B) and runs at a much lower normalized throughput (6.39 versus 102.82 tokens/s in A100-equivalent units). Despite requiring far fewer output tokens per response), its end-to-end verification time on MATH500 (290.30 s) is therefore comparable to that of Prover@16 (324.31 s), rather than dramatically faster.

Moreover, unlike the Lean-based verifier, the LLM-as-a-judge approach provides no formal guarantees and yields a purely heuristic natural-language scoring signal, which is more susceptible to inconsistency and reward hacking. Our empirical results show that the formally verified Prover@16 offers substantially higher reward fidelity and more stable training dynamics for JURY-RL. Taking both wall-clock runtime and supervision quality into account, we view its additional token cost as a practical and acceptable investment in verifier quality.

Table 6: Performance, token cost, and runtime analysis of Lean Verifier and LLM-as-a-judge on MATH500.

| Setting | Model Size | MATH500 | | | Token Costs | Runtime | |
|---|---|---|---|---|---|---|---|
| | | ACC | TPR | F1-Score | Avg Tokens per Response | Eqv. Speed (tokens/s) | Time Cost (s) |
| Prover@16 | 32B | 87.6 | 87.6 | 93.4 | 5,858 | 102.82 | 324.31 |
| Prover@32 | 32B | 89.1 | 89.1 | 94.2 | 6,607 | 102.82 | 506.36 |
| Prover@64 | 32B | 91.1 | 91.1 | 95.3 | 6,435 | 102.82 | 655.80 |
| LLM-as-a-judge (MV@4) | 72B | 87.2 | 87.2 | 93.2 | 464 | 6.39 | 290.30 |

## C.5 PROMPT

We list all the three models' prompts below for reference.

---

**Autoformalizer Prompt**

Please autoformalize the following natural language problem statement in Lean 4.
Use the following theorem name: test_problem
The natural language statement is:
{informal_statement_content}.
Think before you provide the lean statement.

---

**Consistency-Checker System Prompt**

Your role is a Lean4 expert, please help me check consistency between natural language expression and its Lean4 proof statement.
Guidelines for Consistency Checking:
Goal:
Determine if the Lean theorem statement is an exact and faithful formalization of the mathematical problem.
Do not evaluate or consider the answer or the proof. Your sole task is to verify the correctness of the formalization.
Evaluation Stages (All required):
1. Math Assertion Analysis
Identify all structurally and semantically relevant components of the mathematical problem, including variables, types, quantifiers, constraints, logic structure, conclusion, and so on. The analysis should be based on the actual content of the text.
2. Lean Statement Analysis (ignore proof part)
Extract all structurally and semantically relevant components from the Lean statement, including variables, types, conditions, quantifiers, constraints, the final claim, and so on. The analysis should reflect the actual content present in the Lean code.
3. Comparative Verification

---

Check for exact correspondence between the math and Lean statements; you may refer to aspects like:
- Semantic alignment, logic structure, and quantifier correctness.
- Preservation of constraints and boundary assumptions.
- Accurate typing and use of variables.
- Syntactic validity and proper Lean usage (free from errors).
- Use of symbols and constructs without semantic drift.
- No missing elements, no unjustified additions, and no automatic corrections or completions.
4. Final Judgement
Based solely on the above analysis, judge whether the Lean statement is a correct and exact formalization of the mathematical problem.
5. Accuracy Confirmation
If correct: clearly confirm why all elements match.
If incorrect: list all mismatches and explain how each one affects correctness.
Note: While the analysis may be broad and open to interpreting all relevant features, the final judgment must be based only on what is explicitly and formally expressed in the Lean statement.
**Do not consider or assess any part of the proof. Your judgment should be entirely about the accuracy of the statement formalization.**
You should present the results following the format:
Input:
The Natural Language Statement:
A math problem and its answer (no proof).
The Formal Statement in Lean4:
```lean
A Lean 4 theorem statement formalizing the problem. Proof is intentionally omitted (e.g., sorry).
```
Output Format:
Return exactly one xml object
<comments>
Your brief analysis:
Math Assertion Analysis: [. . . ]
Lean Statement Analysis (Proof Ignored): [. . . ]
Comparative Verification: [. . . ]
Conclusion: [. . . ]
Accuracy Confirmation: [. . . match confirmation or list of discrepancies. . . ]
</comments>
<consistency>[Correct/Incorrect]</consistency>

---

### Consistency-Checker User Prompt

Input Data:
The Natural Language Statement:
{informal_prefix}
The Formal Statement in Lean4:
```lean
{formal_statement}
```

---

### Prover Prompt

Complete the following Lean 4 code:
```lean4
{formal_statement}
```
Before producing the Lean 4 code to formally prove the given theorem, provide a detailed proof plan outlining the main proof steps and strategies.

> The plan should highlight key ideas, intermediate lemmas, and proof structures that will guide the construction of the final formal proof.

## D  BASELINE DETAILS

- **Majority Voting (MV).** A self-supervised consensus reward (Shafayat et al., 2025). For each problem, we generate $G$ rollouts. A rollout $y_i$ receives reward 1 if its extracted answer $\text{ans}(y_i)$ matches the majority answer among the $G$ rollouts, and 0 otherwise. This directly reinforces popular answers regardless of correctness.

- **Self-Certainty.** A confidence-derived signal (Zhao et al., 2025b). The reward is computed from the log-probabilities of the tokens composing the final answer (as extracted by $\text{ans}(\cdot)$); higher cumulative log-probability indicates greater model certainty and yields a higher reward.

- **Entropy Minimization.** A low-entropy proxy (Prabhudesai et al., 2025). The reward is inversely related to the policy's output entropy for the final-answer tokens, encouraging more deterministic, high-confidence predictions.

- **CoReward.** Contrastive agreement over input-level variants (Zhang et al., 2025). For each training question, CoReward generates semantically equivalent paraphrased versions, samples multiple rollouts for both the original and paraphrased questions, and aggregates their answers via majority voting to obtain pseudo-consensus labels. These majority-voted answers are then used in a cross-over manner as rewards for policy optimization on both sides.

- **Ground Truth (GT).** The ground-truth supervised oracle baseline. Using human-annotated labels, a rollout receives 1 if its extracted answer matches the ground truth and 0 otherwise. We train this baseline with the same GRPO objective (Shao et al., 2024) as our method for a fair comparison.

- **LLM-as-a-Judge.** An external-judge paradigm (Pang et al., 2023; Zhao et al., 2025a). We employ `qwen-2.5-72b-instruct` as the judge. It assesses the reasoning process and the final answer of each rollout; its evaluation (numeric score or a binary correct/incorrect label) is used as the reward for the RL update. The prompt for it is shown below.

- **LLM-KD (Knowledge Distillation).** Teacher–student distillation has been used in Label-Free RLVR (Zhao et al., 2025a). We use `qwen-2.5-72b-instruct` to produce an answer for each problem and treat it as a pseudo label. The policy model is trained to align its outputs with these machine-generated references.

---

**LLM-as-a-Judge prompt**

You are a professional math QA pair evaluator. Your sole task is to determine whether a given mathematical question and answer are correctly matched. First explain your reasoning, then end your response with your final judgment: True or False.

---

## E  ADDITIONAL EXPERIMENTAL DETAILS

Detailed training and testing settings are provided in Table 7.

## F  BENCHMARK AND METRIC DETAILS

This section details the specific metrics and software frameworks used for each benchmark mentioned in the main text.

- **AIME24/25, MATH500, GSM8K**: We report **avg@16** accuracy for **AIME24/25**, computed by averaging correctness over 16 independently sampled solutions per problem, and **avg@4** accuracy for **MATH500** and **GSM8K**. All benchmarks are evaluated using the `lighteval` framework: https://github.com/huggingface/lighteval

Table 7: Reinforcement learning training hyperparameters. This configuration is consistently applied across all experiments to ensure a fair comparison.

| Hyperparameter | Value |
|---|---|
| *Training Configuration* | |
| Train Batch size (Number of Sampled Questions) | 128 |
| Rollouts per problem ($G$) | 8 |
| Max prompt length | 512 |
| Max new tokens | 3072 |
| Training epoch | 6 |
| *Optimizer Parameters (AdamW)* | |
| Learning rate | $3 \times 10^{-6}$ |
| $\beta_1$ | 0.9 |
| $\beta_2$ | 0.999 |
| $\epsilon$ | $10^{-8}$ |
| Warmup style | Cosine |
| Warmup steps ratio | 0.1 |
| *GRPO Algorithm Parameters* | |
| KL loss coefficient ($\beta$) | 0.005 |
| clip ratio ($\epsilon$) | 0.2 |
| *Generation Parameters* | |
| Training temperature | 1.0 |
| Evaluation temperature | 0.8 |
| top-p | 0.95 |

- **AMC**: We report **avg@8** accuracy. This is evaluated using the `ttrl` implementation: `https://github.com/ruixin31/Spurious_Rewards/tree/main/code/ttrl`

- **LiveCodeBench**: We report **avg@5** accuracy. This is evaluated using the official library: `https://github.com/LiveCodeBench/LiveCodeBench`

- **CRUX**: We report **avg@5** accuracy. This is evaluated using the `ZeroEval` framework: `https://github.com/WildEval/ZeroEval`

- **MMLU-Pro, IFEval**: We report **pass@1** accuracy. This is evaluated using the `lm-evaluation-harness`: `https://github.com/EleutherAI/lm-evaluation-harness`

## G  FURTHER ANALYSIS

### G.1  PASS@K RESULTS

**Main Results across Benchmarks.**  This appendix complements The Diversity Analysis of JURY-RL in Section 5.3. The complete numerical results are listed in Table 8 for easy cross-reference. The pass@k results presented in Table 8 unequivocally demonstrate the effectiveness of our JURY-RL framework. Across all three backbone models, JURY-RL not only surpasses every label-free baseline but also consistently outperforms the strong supervised GT-Reward baseline, which is trained directly on ground-truth answers.

**Ablation Study on Fallback Rewards.**  This appendix complements Ablation Studies of ResZero in Section 5.3. The complete numerical results are listed in Table 9 for easy cross-reference with Table 3. Table 9 demonstrates that our proposed ResZero fallback reward consistently achieves the highest average performance.

Table 8: Pass@k results (%) of *RL performance* comparison on math reasoning benchmarks.

| Methods | AIME24 pass@16 | AIME25 pass@16 | MATH500 pass@4 | GSM8K pass@4 | AMC pass@8 | Average |
|---|---|---|---|---|---|---|
| *Qwen3-1.7B-Base* | | | | | | |
| Before RL | 16.67 | 13.33 | 72.0 | 89.31 | 56.63 | 49.59 |
| GT-Reward | 26.67 | 16.67 | 82.0 | 92.42 | 59.04 | 55.36 |
| LLM-KD | 20.00 | 16.67 | 82.2 | 93.25 | 55.42 | 53.51 |
| Entropy | 20.00 | 23.33 | 81.2 | 90.75 | 59.04 | 54.86 |
| Self-Certainty | 23.33 | 26.67 | 77.2 | 90.83 | 57.83 | 55.17 |
| Majority-Voting | 23.33 | 16.67 | 77.2 | 92.34 | 53.01 | 52.51 |
| CoReward | 26.67 | 16.67 | 77.4 | 92.04 | 54.22 | 53.40 |
| LLM-as-a-Judge | 16.67 | 13.33 | 74.6 | 91.74 | 55.42 | 50.35 |
| **JURY-RL (Ours)** | 30.00 | 30.00 | 82.0 | 92.42 | 62.65 | 59.41 |
| *Llama-3.2-3B-Instruct* | | | | | | |
| Before RL | 23.33 | 6.67 | 65.0 | 86.88 | 49.40 | 46.26 |
| GT-Reward | 23.33 | 6.67 | 63.8 | 90.14 | 43.37 | 45.46 |
| LLM-KD | 23.33 | 3.33 | 65.0 | 89.99 | 49.40 | 46.21 |
| Entropy | 16.67 | 3.33 | 58.8 | 84.46 | 50.60 | 42.77 |
| Self-Certainty | 13.33 | 10.00 | 58.0 | 84.69 | 44.58 | 42.12 |
| Majority-Voting | 16.67 | 0.00 | 63.6 | 90.98 | 39.76 | 42.20 |
| CoReward | 23.33 | 0.00 | 60.4 | 90.45 | 43.37 | 43.51 |
| LLM-as-a-Judge | 26.67 | 10.00 | 62.2 | 89.84 | 40.96 | 45.93 |
| **JURY-RL (Ours)** | 30.00 | 3.33 | 64.8 | 90.07 | 54.22 | 48.48 |
| *Qwen2.5-7B* | | | | | | |
| Before RL | 30.00 | 20.00 | 76.2 | 93.78 | 63.86 | 56.77 |
| GT-Reward | 33.33 | 30.00 | 85.2 | 95.22 | 68.67 | 62.48 |
| LLM-KD | 33.33 | 33.33 | 87.8 | 95.75 | 67.47 | 63.54 |
| Entropy | 30.00 | 36.67 | 85.2 | 93.78 | 67.47 | 62.62 |
| Self-Certainty | 23.33 | 30.00 | 84.6 | 93.86 | 69.88 | 60.33 |
| Majority-Voting | 33.33 | 30.00 | 85.4 | 95.00 | 63.86 | 61.52 |
| CoReward | 30.00 | 26.67 | 83.8 | 95.60 | 63.86 | 59.99 |
| LLM-as-a-Judge | 20.00 | 13.33 | 82.2 | 95.38 | 59.04 | 53.99 |
| **JURY-RL (Ours)** | 36.67 | 30.00 | 86.6 | 95.83 | 71.08 | 64.04 |

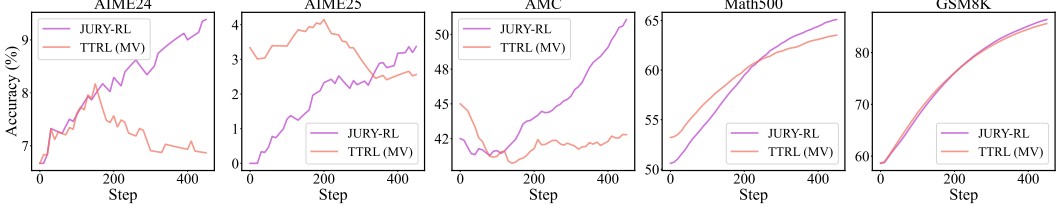

Figure 6: **Test-Time Training Dynamics on Qwen2.5-7B.** We plot exponentially moving-averaged (EMA) validation accuracy for Jury-RL (purple) and majority-voting TTRL (orange) over 400 test-time training steps. Jury-RL exhibits more stable and continued improvement on challenging benchmarks.

## G.2 TEST-TIME TRAINING RESULTS

While our main experiments focus on offline RLVR, recent work has emphasized *test-time* reinforcement learning (TTRL) (Zuo et al., 2025), where the model is adapted on the evaluation distribution itself. To examine whether the conclusions in §5.3 also hold in this regime, we instantiate a TTRL variant of Jury-RL and compare it with the majority-vote (MV) reward used in the public `ttrl` implementation. For each benchmark (AIME24, AIME25, AMC23, MATH500 and GSM8K), we

Table 9: Ablation results (Pass@k) for the proposed ResZero reward ($\delta = 0$) on math reasoning benchmark.

| Methods | AIME24 pass@16 | AIME25 pass@16 | MATH500 pass@4 | GSM8K pass@4 | AMC pass@8 | Average |
|---|---|---|---|---|---|---|
| *Qwen3-1.7B-Base* | | | | | | |
| GT-Reward | 26.67 | 16.67 | 82.0 | 92.42 | 59.04 | 55.36 |
| Majority-Voting | 23.33 | 16.67 | 77.2 | 92.34 | 53.01 | 52.51 |
| Proof-Gate + Zero Reward | 23.33 | 6.67 | 70.2 | 92.65 | 46.99 | 47.97 |
| Proof-Gate + Random Reward | 20.00 | 6.67 | 78.0 | 92.34 | 54.22 | 50.25 |
| Proof-Gate + MV Reward | 0.00 | 0.00 | 48.8 | 92.19 | 13.25 | 30.85 |
| **JURY-RL (Proof-Gate + ResZero)** | 30.00 | 30.00 | 82.0 | 92.42 | 62.65 | 59.41 |
| *Llama-3.2-3B-Instruct* | | | | | | |
| GT-Reward | 23.33 | 6.67 | 63.8 | 90.14 | 43.37 | 45.46 |
| Majority-Voting | 16.67 | 0.00 | 63.6 | 90.98 | 39.76 | 42.20 |
| Proof-Gate + Zero Reward | 20.00 | 6.67 | 64.0 | 89.92 | 40.96 | 44.31 |
| Proof-Gate + Random Reward | 23.33 | 6.67 | 62.6 | 88.32 | 46.99 | 45.58 |
| Proof-Gate + MV Reward | 20.00 | 0.00 | 62.2 | 89.76 | 44.58 | 43.31 |
| **JURY-RL (Proof-Gate + ResZero)** | 30.00 | 3.33 | 64.8 | 90.07 | 54.22 | 48.48 |
| *Qwen2.5-7B* | | | | | | |
| GT-Reward | 33.33 | 30.00 | 85.2 | 95.22 | 68.67 | 62.48 |
| Majority-Voting | 33.33 | 30.00 | 85.4 | 95.00 | 63.86 | 61.52 |
| Proof-Gate + Zero Reward | 30.00 | 33.33 | 85.6 | 95.38 | 62.65 | 61.39 |
| Proof-Gate + Random Reward | 33.33 | 16.67 | 79.6 | 93.63 | 63.86 | 57.42 |
| Proof-Gate + MV Reward | 0.00 | 0.00 | 58.2 | 95.60 | 19.28 | 34.62 |
| **JURY-RL (Proof-Gate + ResZero)** | 36.67 | 30.00 | 86.6 | 95.83 | 71.08 | 64.04 |

treat the validation split as the TTRL environment, start from the same base policy, and run GRPO for 30 epochs with the hyperparameters in Table 7. The only difference between the two curves in Figure 6 is the reward: MV vs. our ResZero fallback.

The dynamics on AIME24 and AMC highlight the brittleness of MV in the test-time setting: the validation reward first increases slightly and then degrades, eventually falling below the initial performance, whereas JURY-RL continues to make steady progress with a smoother learning curve, consistent with our analysis in Appendix A.1 that spurious consensus under MV drives entropy collapse and hurts generalization. On AIME25, MATH500, and GSM8K the two methods follow similar learning trajectories, but JURY-RL consistently finishes more than one accuracy point ahead of MV; the curves on MATH500 and GSM8K are close yet JURY-RL ends higher, and on AIME25 it attains a more stable plateau.

The dynamics on AIME24 and AMC highlight the brittleness of MV in the test-time setting: the validation reward first increases slightly and then degrades, eventually falling below the initial performance, whereas JURY-RL continues to make steady progress with a smoother learning curve, consistent with our analysis in Appendix A.1 that spurious consensus under MV drives entropy collapse and hurts generalization. For AIME25 in particular, the first points in Figure 6 correspond to 0/30 (JURY-RL) and 1/30 (MV) solved problems; since both runs start from the same base model, this small gap is due to stochasticity in GRPO rather than an inherent disadvantage of JURY-RL. As training proceeds, JURY-RL attains a more stable plateau on AIME25 and still ends slightly ahead of MV, and on MATH500 and GSM8K the two curves remain close but JURY-RL consistently finishes higher.

### G.3 DEEPER DIVE INTO VERIFIER IMPERFECTIONS

While a formal verifier is theoretically sound, its practical application is imperfect. We analyze how the number of verification attempts, $k$ in a $pass@k$ setting, affects the quality of the reward signal. As shown in Figure 7, the Lean verifier exhibits a highly desirable trade-off.

The **precision** (top-left panel) remains consistently high, around 85%. It is crucial to note that this precision gap from a perfect 100% is not a flaw in the Lean prover's core logic, which is formally sound. Instead, it primarily stems from imperfections in upstream components like **auto-formalization and consistency checks**, which translate the problem into a verifiable format. De-

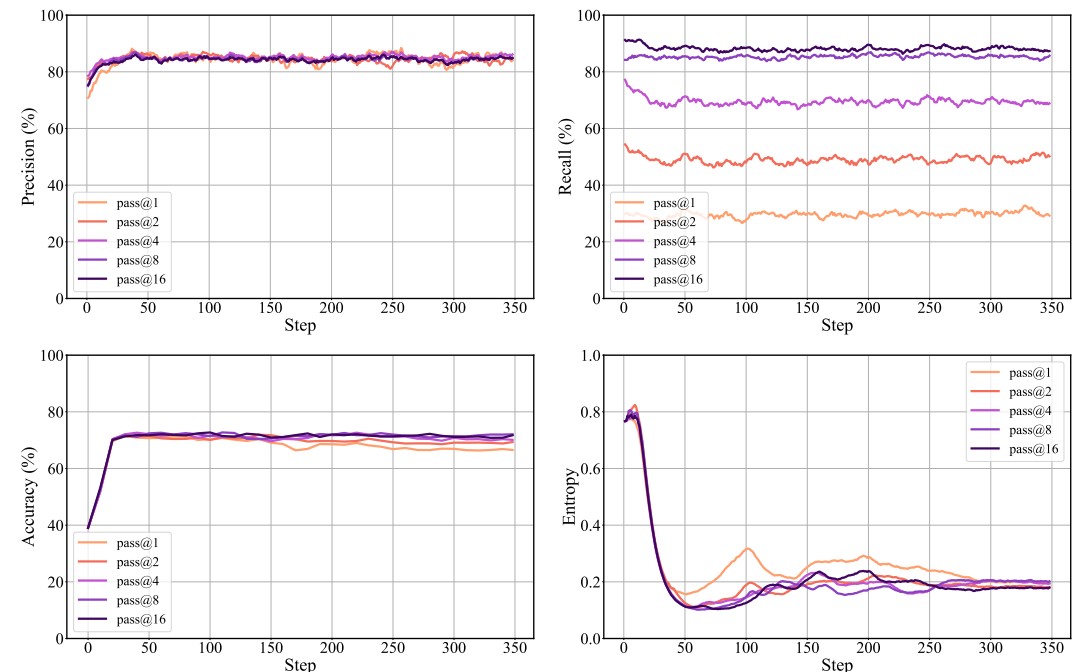

Figure 7: Training dynamics of precision, recall, validation accuracy, and training entropy under different Lean $pass@k$ verification settings.

spite this, the signal's high fidelity is vital for preventing reward hacking and ensuring the model learns from genuinely correct examples.

In contrast, the **recall** (top-right panel) shows a clear dependency on $k$. With a single attempt ($pass@1$), the recall is modest (around 30%), but it steadily increases with more attempts, reaching nearly 90% at $pass@16$. This indicates that while a single verification attempt might fail to prove a correct answer (e.g., due to search limits), multiple attempts significantly increase the chance of success, thereby improving the richness of the training signal.

Crucially, despite the wide variance in recall, the final **validation accuracy** (bottom-left) and **training entropy** (bottom-right) converge to similar stable states across different values of $k$. This suggests that the high precision of the verifier's signal is the dominant factor for successful and stable training. This behavior stands in stark contrast to an LLM-as-a-Judge. **Mechanistically**, an LLM-judge's potential for error is inherent to its probabilistic and opaque reasoning process, making it fundamentally prone to biases, format gaming, and confident mistakes. Its reward signal is therefore inherently noisier and less reliable. The imperfections in our pipeline, however, are largely confined to the modular pre-processing steps, leaving the core judgment by the Lean prover itself principled and trustworthy.

**Effect of top-$k$ verification on performance.** To quantify how many candidates actually need to be verified, we compare JURY-RL training runs on Qwen2.5-7B that verify only the top-1, top-2, or top-3 majority-voting candidates selected by the policy, and report accuracy on the MATH5000 validation set in Figure 8. All three curves reach a similar plateau around 72% accuracy, and they remain almost indistinguishable throughout training, with final differences well within random fluctuation. This indicates that almost all useful learning signal is already contained in the highest-probability candidate, and additional verified candidates mainly increase computational cost without improving final accuracy. Conse-

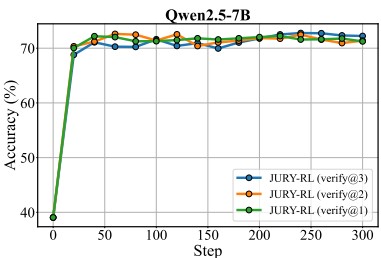

Figure 8: Accuracy on MATH5000 Validation set over training steps.

quently, we adopt the more efficient top@1 verify setting as our default choice in all main experiments.

## H  CASE STUDY

To illustrate our "Proofs Dispose" principle, we present a case study comparing an LLM-as-a-Judge with a Lean formal verifier. This case highlights a core problem with LLM judges: their reliability is limited by their own reasoning. The first example shows that an LLM judge can reach a wrong conclusion due to flawed reasoning, which generates an incorrect training signal. In contrast, once the autoformalization and consistency checks are passed, a Lean proof has the property of zero false positives. As shown in the second example, it provides a definitive and highly reliable signal through formal logic. This comparison explains why in JURY-RL, we use proofs, not an LLM, to dispose the final reward.

---

**Question**

```
Let $x$ and $y$ be positive real numbers such that $3x + 4y < 72.$
    ↪ Find the maximum value of xy (72 - 3x - 4y).
```

---

**Answer**

```
1152
```

---

**LLM-as-a-Judge Output (Judge Wrong)**

```
To evaluate whether the given answer of 1152 is correct for the
    ↪ problem, we need to find the maximum value of the function \(
    ↪ f(x, y) = xy (72 - 3x - 4y) \) subject to the constraint \( 3x
    ↪ + 4y < 72 \).

First, let's rewrite the function:
\[ f(x, y) = xy (72 - 3x - 4y). \]

We can use the method of Lagrange multipliers or analyze the function
    ↪ directly. However, a simpler approach is to use symmetry and
    ↪ calculus.

Consider the function \( g(t) = t(72 - 3t - 4t) = t(72 - 7t) = 72t -
    ↪ 7t^2 \). This is a quadratic function in terms of \( t \),
    ↪ where \( t = x + y \). The maximum value of this quadratic
    ↪ function occurs at the vertex, which is given by:
\[ t = -\frac{b}{2a} = -\frac{72}{-14} = \frac{72}{14} =
    ↪ \frac{36}{7}. \]

Now, we need to find \( x \) and \( y \) such that \( x + y =
    ↪ \frac{36}{7} \) and \( 3x + 4y \) is minimized while still
    ↪ satisfying \( 3x + 4y < 72 \).

Given the symmetry and the nature of the problem, we can assume \( x
    ↪ = y \). Then:
\[ 2x = \frac{36}{7} \implies x = \frac{18}{7}. \]
\[ 3x + 4y = 3\left(\frac{18}{7}\right) + 4\left(\frac{18}{7}\right)
    ↪ = 7\left(\frac{18}{7}\right) = 18. \]

Substituting \( x = y = \frac{18}{7} \) into the function:
\[ f\left(\frac{18}{7}, \frac{18}{7}\right) =
    ↪ \left(\frac{18}{7}\right)\left(\frac{18}{7}\right)(72 -
    ↪ 3\left(\frac{18}{7}\right) - 4\left(\frac{18}{7}\right)) \]
\[ = \left(\frac{18}{7}\right)^2 (72 - 7\left(\frac{18}{7}\right)) \]
```

---

```
\[ = \left(\frac{18}{7}\right)^2 (72 - 18) \]
\[ = \left(\frac{18}{7}\right)^2 \cdot 54 \]
\[ = \frac{324}{49} \cdot 54 \]
\[ = \frac{17496}{49} \]
\[ = 357.0612244898. \]

This value does not match 1152. Therefore, the given answer of 1152
    ↪ is incorrect.

Final judgment: False.
```

**Lean Prover Output (Prove Success)**

```
import Mathlib
import Aesop

set_option maxHeartbeats 0

open BigOperators Real Nat Topology Rat

theorem test_problem :
  sSup {z : ℝ | ∃ x y : ℝ, x > 0 ∧ y > 0 ∧ 3*x + 4*y < 72 ∧ z = x *
    ↪ y * (72 - 3*x - 4*y)} = 1152 := by
  have h₁ : (1152 : ℝ) ∈ {z : ℝ | ∃ x y : ℝ, x > 0 ∧ y > 0 ∧ 3*x +
    ↪ 4*y < 72 ∧ z = x * y * (72 - 3*x - 4*y)} := by
    refine' ⟨8, 6, by norm_num, by norm_num, by norm_num, _⟩
    <;> norm_num
    <;> ring_nf
    <;> norm_num

  have h₂ : ∀ (z : ℝ), z ∈ {z : ℝ | ∃ x y : ℝ, x > 0 ∧ y > 0 ∧ 3*x +
    ↪ 4*y < 72 ∧ z = x * y * (72 - 3*x - 4*y)} → z ≤ (1152 : ℝ) :=
    ↪ by
    intro z hz
    rcases hz with ⟨x, y, hx, hy, hxy, rfl⟩
    have h₃ : 0 < x := hx
    have h₄ : 0 < y := hy
    have h₅ : 3 * x + 4 * y < 72 := hxy
    have h₆ : 0 < 72 - 3 * x - 4 * y := by linarith
    have h₇ : 0 < 3 * x := by positivity
    have h₈ : 0 < 4 * y := by positivity
    have h₉ : 0 < (3 * x) * (4 * y) := by positivity
    have h₁₀ : 0 < (3 * x) * (4 * y) * (72 - 3 * x - 4 * y) := by
    ↪ positivity
    -- Use AM-GM inequality to show that x * y * (72 - 3 * x - 4 * y)
    ↪ ≤ 1152
    have h₁₁ : (3 * x + 4 * y + (72 - 3 * x - 4 * y)) ^ 3 ≥ 27 * (3 *
    ↪ x) * (4 * y) * (72 - 3 * x - 4 * y) := by
      nlinarith [sq_nonneg (3 * x - 4 * y), sq_nonneg (3 * x - (72 -
    ↪ 3 * x - 4 * y)), sq_nonneg (4 * y - (72 - 3 * x - 4 * y)),
        mul_nonneg h₇.le h₈.le, mul_nonneg h₇.le h₆.le, mul_nonneg
    ↪ h₈.le h₆.le]
    -- Simplify the inequality to get the desired result
    have h₁₂ : (72 : ℝ) ^ 3 ≥ 27 * (3 * x) * (4 * y) * (72 - 3 * x -
    ↪ 4 * y) := by
      have h₁₃ : (3 * x + 4 * y + (72 - 3 * x - 4 * y)) = 72 := by
    ↪ ring
      rw [h₁₃] at h₁₁
      linarith
    have h₁₃ : x * y * (72 - 3 * x - 4 * y) ≤ 1152 := by
      have h₁₄ : (72 : ℝ) ^ 3 = 373248 := by norm_num
```

```
      have h₁₅ : 27 * (3 * x) * (4 * y) * (72 − 3 * x − 4 * y) = 324
    ↪ * (x * y * (72 − 3 * x − 4 * y)) := by
        ring
      rw [h₁₄] at h₁₂
      rw [h₁₅] at h₁₂
      nlinarith
    linarith

  have h₃ : sSup {z : ℝ | ∃ x y : ℝ, x > 0 ∧ y > 0 ∧ 3*x + 4*y < 72 ∧
    ↪  z = x * y * (72 − 3*x − 4*y)} = 1152 := by
    have h₄ : IsLUB {z : ℝ | ∃ x y : ℝ, x > 0 ∧ y > 0 ∧ 3*x + 4*y <
    ↪ 72 ∧ z = x * y * (72 − 3*x − 4*y)} 1152 := by
      refine' ⟨fun z hz => h₂ z hz, fun y hy => _⟩
      -- We need to show that 1152 is the least upper bound.
      -- Since 1152 is in the set, any upper bound y must be at least
    ↪ 1152.
      have h₅ : (1152 : ℝ) ∈ {z : ℝ | ∃ x y : ℝ, x > 0 ∧ y > 0 ∧ 3*x
    ↪ + 4*y < 72 ∧ z = x * y * (72 − 3*x − 4*y)} := h₁
      have h₆ : (1152 : ℝ) ≤ y := by
        apply hy
        exact h₅
      linarith
    -- The supremum of the set is 1152 because it is the least upper
    ↪ bound.
    have h₅ : sSup {z : ℝ | ∃ x y : ℝ, x > 0 ∧ y > 0 ∧ 3*x + 4*y <
    ↪ 72 ∧ z = x * y * (72 − 3*x − 4*y)} = 1152 := by
      apply le_antisymm
      · -- Show that the supremum is less than or equal to 1152.
        apply csSup_le
        · -- The set is nonempty.
          exact ⟨1152, h₁⟩
        · -- Every element in the set is less than or equal to 1152.
          intro z hz
          exact h₂ z hz
      · -- Show that 1152 is less than or equal to the supremum.
        apply le_csSup
        · -- The set is bounded above.
          use 1152
          intro z hz
          exact h₂ z hz
        · -- 1152 is in the set.
          exact h₁
    exact h₅

  exact h₃
```

