# OpenReview forum: "JURY-RL: Votes Propose, Proofs Dispose for Label-Free RLVR"
_ICLR.cc/2026/Conference — Submitted to ICLR 2026_

### Official Review · Reviewer_kMKV · 2025-10-29

**Soundness:** 3
**Presentation:** 3
**Contribution:** 3
**Rating:** 4
**Confidence:** 3

**Summary:**

This paper introduces JURY-RL, a framework for label-free RLVR. A formal Lean theorem prover is used to dispose the reward by verifying whether the proposed answer is provably correct. If verification succeeds, the model is rewarded for correctness. When verification is inconclusive, the paper proposes a ResZero fallback reward, which maintains a zero-mean, variance-preserving gradient signal to stabilize rewards. Experiments across different benchmarks show that JURY-RL improves stability over self-reward and LLM-judge baselines.

**Strengths:**

1. Using Lean as a proof gate bridges RLVR with formal methods.

2. The proposed zero-mean fallback reward stabilizes optimization even when verification fails.

3. The experiments are broad, covering multiple backbones and domains. JURY-RL consistently outperforms other label-free baselines.

**Weaknesses:**

1. The proposed pipeline uses a 32B auto-formalizer (8 candidates), a 32B consistency checker, and a 32B prover (16 proofs), making each rollout more expensive than LLM-as-Judge. Yet the performance gain is less than 3 points on average, questioning the cost–benefit balance.

2. Although Lean is rigorous, the upstream translation and proof-generation models may introduce false positives or negatives. The paper does not quantify conversion accuracy, proof validity rates, or error propagation.

3. The impact of each module (formalizer, checker, prover) is not isolated, making it unclear which contributes to the observed improvements.

**Questions:**

1. What is the total verification cost (e.g. GPU hours/ LLM calls per rollout) compared to LLM-as-Judge? Can you provide a more detailed cost-benefit curve or analysis, e.g., the impact of different autoformalization/ proof candidate number on RL performance?

2. Can you report the accuracy of auto-formalization and model generated Lean proofs?

3. Can you justify why your method performance is better than the ground truth answer setting in table 1 and 2?

I will increase the score if all the questions are well-justified.

---

> ### Author Response · Authors · 2025-12-01
> **Response to Reviewer kMKV (Part 1/3)**
>
> We thank the reviewer for the insightful and constructive feedback. We appreciate your acknowledgment of the novelty of bridging RLVR with formal methods and the stability of our ResZero reward. We understand your concerns regarding the cost-benefit balance and the reliability of the verification pipeline. Below, we address each question and describe the new experiments and analyses added to the revised manuscript.
>
> ### 1. Response to W1 & Q1: Cost-Benefit Analysis & Verification Cost
>
> **High-Value Gains in avg@k and Pass@k.**
> The reviewer noted that the “performance gain is less than 3 points on average.” While the absolute number may look small in isolation, in our setting a **+3-point avg@k improvement** is already substantial: JURY-RL consistently opens a gap over all other **label-free** baselines in Table 1, and its avg@k performance becomes comparable to the GT-Reward oracle, which relies on expensive ground-truth labels.
> Moreover, avg@k underestimates the benefit in realistic deployments. For reasoning systems that are actually used with sampling or search, **Pass@k** is the more relevant indicator of the model’s latent problem-solving ability. As shown in the new **Table 2 (main text)**, JURY-RL provides much larger gains in Pass@k than in Pass@1:
> * On **Qwen2.5-7B**, JURY-RL improves **Pass@k by +10.05 points** over LLM-as-a-Judge (53.99% → 64.04%) and still exceeds the GT-Reward oracle by +1.56 points.
> * On **Qwen3-1.7B**, JURY-RL improves **Pass@k by +9.06 points** over LLM-as-a-Judge (50.35% → 59.41%) and outperforms GT-Reward by +4.05 points.
> These results indicate that JURY-RL not only achieves non-trivial gains on avg@k, but also delivers **over 9–10 absolute points** improvement in the practically important Pass@k metric.
>
>
> **Cost Analysis: verification strategy and wall-clock overhead.**
>
> 1. **Verifying only the top-1 candidate is enough.**
>    To quantify how many candidates actually need to be sent to the verifier, we ran JURY-RL on Qwen2.5-7B with three settings that verify only the top-1, top-2, or top-3 majority-voting answers selected by the policy, and tracked accuracy on the MATH5000 validation set (new **Figure 8**). All three curves rapidly rise to and then stay around **≈72%** accuracy; they remain almost indistinguishable throughout training, with final differences well within run-to-run randomness.
>    This shows that almost all useful learning signal is already contained in the highest-probability candidate. Verifying additional majority-vote answers mostly increases computational cost without improving accuracy. Therefore, in all main experiments we adopt the **top-1-only verification** setting, so each problem triggers at most a single Lean verification in the inner RL loop.
>
> 2. **Efficient prover configuration vs. LLM-as-a-Judge.**
>    Appendix C.3 further analyzes the trade-off between verifier strength and runtime. On MATH500, a stronger Lean configuration **Prover@64** attains the best accuracy but requires **655.8 s** end-to-end, whereas **Prover@16** achieves a very similar **87.6% accuracy** with roughly **half the time (324.3 s)**.  We therefore use **Prover@16** as the default Lean-based verifier.
>    For comparison, a Qwen2.5-72B-instruct **LLM-as-a-Judge (mv@4)** has comparable accuracy (87.2%) but needs a much larger model (72B vs. 32B) and runs at a far lower normalized throughput (6.39 vs. 102.82 tokens/s). As a result, its end-to-end verification time on MATH500 (**290.3 s**) is **comparable rather than dramatically smaller** than Prover@16 (**324.3 s**).
>    In other words, once we factor in model size and throughput, Lean-based verification is not orders of magnitude slower than an LLM judge, while providing formally sound rewards instead of heuristic natural-language scores.
>
> 3. **Early stopping and pass@k evaluation.**
>    Both during training-time verification and evaluation-time pass@k metrics, the Lean pipeline employs **early stopping**: we attempt auto-formalization and proof up to k times but terminate immediately once a single proof succeeds. Since we always verify the majority-vote answer, which already has high empirical precision, the prover typically succeeds within the first few attempts rather than exhausting all k trials.  This significantly reduces the *effective* cost of using higher pass@k thresholds while still enjoying their recall benefits (Figure 7).
>
> (Response continued in Part 2/3)

---

> > ### Author Response · Authors · 2025-12-01
> > **Response to Reviewer kMKV (Part 2/3)**
> >
> > (Continued from Part 2/3)
> >
> > 4. **Cold-start overhead and amortized cost.**
> >    Appendix C.3 also provides a wall-clock breakdown on Qwen2.5-7B. Under our synchronous RL setup, a step with GT-Reward (no external verification) takes about **100 s**. Adding Lean-based verification for the majority-vote answer introduces roughly **+200 s** in a worst-case “all QA pairs unseen” batch, whereas an LLM-as-a-Judge adds about **+80 s**. Thus the initial step times are ≈300 s for JURY-RL and ≈180 s for LLM-as-a-Judge.
> >    However, this is a pessimistic upper bound. As training progresses, we cache all verification results. When the policy starts to converge, an increasing fraction of batch questions reuse previously verified answers; those hits are served directly from the cache at negligible cost. As shown in Appendix C.3, this caching effect causes the per-step runtime of JURY-RL to gradually approach that of the GT-Reward baseline.
> >
> > In summary, we (i) restrict verification to the top-1 majority candidate per question, (ii) select an efficient Lean configuration (Prover@16) whose runtime is comparable to a large LLM-as-a-Judge once model size and throughput are taken into account, (iii) exploit early stopping for pass@k verification, and (iv) amortize costs over training via caching. These design choices keep the *effective* verification cost moderate, while yielding **>9–10 point improvements in Pass@k** over LLM-as-a-Judge and consistent gains over GT-Reward. We therefore view the additional verifier cost as a practical and well-justified investment in reward fidelity and downstream reliability.
> >
> >
> >
> > ### 2. Response to W2, W3 & Q2: Verifier Component Analysis & Reliability
> >
> > We address the concerns regarding **error quantification (W2/Q2)** and **module isolation (W3)** by evaluating the independent performance of each pipeline component on MATH500 (details added in **Appendix C.3**).
> >
> > * **Autoformalizer (High Coverage):** The module is highly robust, successfully generating syntactically valid and type-correct Lean statements for **95.4%** (477/500) of problems.
> > * **Consistency Checker (High Fidelity):** By isolating this component, we find it effectively filters semantic mismatches with an **F1-score of 91.1%** and **Recall of 94.9%**, ensuring high-quality signal filtration.
> > * **Prover (Sound Guardrail):** Unlike probabilistic LLM judges, the Lean kernel acts as a sound logical guardrail, ensuring a **0% False Positive Rate** for the proof verification step itself.
> >
> > **End-to-End Signal Quality:**
> > Combining these isolated modules, the system achieves a precision of **84.5%**—significantly higher than LLM-as-a-Judge (75.9%). While single-pass recall is limited by search depth, our **Pass@16** strategy boosts effective recall to **88.0%**. For inconclusive cases (e.g., search timeouts), **ResZero** provides a stable fallback, ensuring that limitations in individual modules do not destabilize RL training.

---

> > > ### Author Response · Authors · 2025-12-01
> > > **Response to Reviewer kMKV (Part 3/3)**
> > >
> > > (Continued from Part 2/3)
> > >
> > >
> > > **3. Response to Q3: Performance Analysis vs. Ground Truth (GT) Rewards**
> > >
> > > Our updated results (Table 1 & 2) reveal that while JURY-RL performs comparably to the supervised GT-Reward in average expected accuracy (`avg@k`), it significantly outperforms GT in **solution diversity and latent reasoning capability** (`pass@k`).
> > >
> > > **1. Comparable Avg@k, Superior Pass@k (Table 1 vs. Table 2).**
> > > As shown in **Table 1**, JURY-RL effectively matches the strong supervised GT-Reward baseline on avg@k across benchmarks (e.g., Qwen2.5-7B on MATH500: 75.65% vs. 75.1%). However, **Table 2** highlights a critical advantage: JURY-RL achieves substantially higher **pass@k** scores.
> > > * On **Qwen2.5-7B**, JURY-RL surpasses GT-Reward by **+1.56 points** and LLM-as-a-Judge by **+10.05 points** in pass@k.
> > > * On **Qwen3-1.7B**, the gap is even larger, with JURY-RL outperforming GT-Reward by **+4.05 points** in pass@k.
> > > This indicates that while both methods produce correct answers with similar frequency on average, JURY-RL explores a much broader space of valid solutions.
> > >
> > > **2. Combating Mode Collapse with ResZero.**
> > > The performance difference stems from how the rewards shape the policy. GT-Reward supervises against a static ground-truth string, pushing the policy to collapse onto a single canonical reasoning path (Mode Collapse). In contrast, JURY-RL employs the **ResZero** mechanism, which:
> > > * **Penalizes unverified consensus:** Prevents the model from converging to a single, potentially spurious solution.
> > > * **Redistributes rewards to residuals:** Explicitly encourages the exploration of alternative, minority reasoning paths.
> > > This mechanism maintains high solution diversity throughout training (as evidenced by the unique answer analysis in Figure 2), allowing the model to find correct solutions that supervised training might miss due to overfitting.
> > >
> > > **3. Robust Generalization.**
> > > This diversity translates into robustness. In **Table 1**, JURY-RL consistently achieves the highest performance among label-free methods and matches GT-Reward on out-of-distribution benchmarks like **GSM8K** and **AMC**. By validating logic via Lean rather than matching a reference string, JURY-RL learns generalized reasoning principles rather than dataset-specific formatting, making it a more scalable alternative for label-free scenarios.
> > >
> > > ---
> > >
> > > We believe the added experiments (Tables 1–2; Appendix C.3) effectively resolve concerns regarding cost and reliability. The results confirm that JURY-RL yields substantial **Pass@k gains (>9 points)** and robust generalization comparable to supervised baselines, all within a practical compute budget. We hope these clarifications justify a positive reassessment of our work.

---

### Official Review · Reviewer_a7FE · 2025-10-30

**Soundness:** 3
**Presentation:** 3
**Contribution:** 3
**Rating:** 6
**Confidence:** 3

**Summary:**

The submission proposes a hybrid of LLM-voting and label-free RLVR, where a committee proposes a derivation and then a formal theorem prover verifies it. When the proposed derivation cannot be verified, they fall back to a heuristic reward that achieves zero mean by assigning negative reward to the plurality answer and positive reward to other candidate answers. In this way it is possible to learn even from answers that cannot be verified.

Experiments show that this approach improves significantly over LLM-as-judge, and also outperforms RLVR from ground truth rewards in most cases. Ablations suggest that simply assigning zero reward to unverified derivations does not work, so the proposed approach is justified empirically. Experiments also demonstrate the stability of the method (figure 2), the ability to preserve answer diversity (figure 3), and the impact of the main hyperparameter (figure 4).

**Strengths:**

- The paper tackles an important problem: relaxing the need for human annotation while preserving the validity of the annotations.
- The proposed approach is mathematically sound and appears to be practical to implement
- The motivation of scalability, truth-alignment, and optimization-stability is persuasive
- The paper includes unusually comprehensive experimental validation, including ablations and robustness checks
- The worked example in B.2 is helpful for understanding.

**Weaknesses:**

It would be great to have more formal understanding of the ResZero reward. In particular, I don't have an intuition for why it is a good idea to  penalize the majority and amplifying the residuals in proportion to their frequency. It seems like this could even lead to an oscillatory behavior where two hypotheses alternate as the "majority" and top "residual".

Minor: "Majority" typically refers to >50%, but here I think what is meant is "plurality"

**Questions:**

- How is $\overline{u}$ computed?
- What happens in ResZero when all the samples are different answers? Do you pick a "majority" answer at random?
- Can you please formally express the proposition being proved in A.2?
- Table 2 shows that assigning zero reward to unverified answers is generally less effective than ResZero. What about assigning a zero-mean random reward?
- How are the CIs computed for the averages in tables 1 & 2?

---

> ### Author Response · Authors · 2025-12-01
> **Response to Reviewer a7FE (Part 1/3)**
>
> We thank the reviewer for the thoughtful and positive assessment, and for highlighting both the practical importance of label-free RLVR and the strengths of our empirical validation. We are grateful for the detailed questions on the ResZero reward, which helped us clarify both its intuition and formal properties. In the revised manuscript we have added explanations, a formal proposition, and additional ablations accordingly.
>
>
> ### Response to W1: Intuition and formal understanding of the ResZero reward
>
> **When is the majority penalized?**
> ResZero is only activated in the *failure* case of the verifier. If the majority-vote answer $\hat{a}$ can be formally verified ($\delta = 1$), we use a standard, positive reward for trajectories that support $\hat{a}$, and no penalty is applied to the majority. The majority is penalized *only* when verification fails ($\delta = 0$), i.e., exactly in the regime where we have strong but *unverified* consensus and wish to avoid “being confidently wrong”.
>
> **Why penalize the unverifiable majority and amplify residuals?**
> When verification is inconclusive, we know that the current majority proposal is *not backed* by a formal proof; treating it as if it were correct (as in MV) risks reinforcing a spurious consensus and driving entropy collapse. Our goal in this case is:
>
> * to **reduce** probability mass on the unverifiable consensus, and
> * to **redistribute** that probability toward alternative hypotheses that still have some empirical support in the residual group.
>
> ResZero does exactly this. For a group of $G$ rollouts, let $M$ be the set of trajectories supporting the (unverified) plurality $\hat{a}$ and $R$ be the remaining “residual” trajectories. The majority share is $\alpha = |M| / G$. Within the residual group, we compute a leave-one-out support $z_i$ for each trajectory and subtract the residual mean $\bar{u}$. The term $\alpha (z_i - \bar{u})$ then provides a **zero-mean signal within the residuals**, rewarding residual answers that are more popular than the average minority and penalizing those that are less popular, while all majority trajectories receive a negative offset $-c\alpha$.
>
> Intuitively:
>
> * If the model is **already uncertain** (small $\alpha$), we want only a mild corrective step; the ResZero signal is correspondingly small.
> * If the model is **confident but inconclusive** (large $\alpha$ and $\delta = 0$), we want a stronger push away from the unverifiable consensus; scaling by $\alpha$ makes both the majority penalty and the residual amplification stronger exactly in this regime.
>
> Thus, “penalizing the majority” is not an arbitrary choice: it is a mechanism to *undo* confidence in unverifiable answers, while the residual term uses the empirical structure of the minority votes to guide exploration toward more promising alternatives.
>
> **Why this does not lead to unstable oscillations.**
> The oscillation concern is very reasonable: in principle, two hypotheses could alternate as “majority” and “top residual”. ResZero mitigates this in three ways:
>
> 1. **Zero-mean, bounded signal.**
>    By construction, the group reward always sums to zero, and as analyzed in Appendix A.2 and A.3, the group variance is non-degenerate whenever residual answers are not all identical. This yields a *balanced* update (no runaway drift in expectation) while preserving an informative gradient.
>
> 2. **Damped corrections rather than flips.**
>    Consider two competing, unverifiable answers $A$ and $B$. When $A$ becomes an over-dominant majority and verification fails, all trajectories supporting $A$ receive a negative advantage, while those supporting $B$ receive a positive advantage proportional to their residual support. This *reduces* the probability of $A$ and increases that of $B$, pushing the policy toward a more balanced mixture. If the roles later reverse, the same mechanism acts in the opposite direction, leading to *damped* adjustments around a mixed solution, not unbounded oscillation.
>
> 3. **Interaction with GRPO and KL regularization.**
>    The ResZero rewards are fed into GRPO with ratio clipping and a KL term to the reference policy. These mechanisms further limit step size and prevent large swings in policy probabilities. In other words, ResZero provides a *directional* signal (penalize unverifiable consensus, favor relatively stronger residuals), while GRPO ensures the resulting updates remain stable.
>
> Empirically, Figure 2 directly addresses this concern: training curves with ResZero are smooth and stable, whereas entropy-based and self-certainty baselines exhibit clear collapse. This matches our theoretical analysis in Appendix A: among the fallback strategies (MV, Zero, ResZero), only ResZero yields a corrective yet exploratory update that avoids collapse under GRPO.
>
> We have revised **Appendix A.3 Case 3: ResZero Reward** in the updated manuscript to make this “control” perspective more explicit.
>
> (Response continued in Part 2/3)

---

> > ### Author Response · Authors · 2025-12-01
> > **Response to Reviewer a7FE (Part 2/3)**
> >
> > (Continued from Part 1/3)
> >
> > ### Response to Q1: How is $\bar{u}$ (the residual mean) computed?
> >
> > Thank you for pointing out that the definition of the residual mean $\bar{u}$ in Eq. (5) was not fully explicit. For a given question, we proceed as follows:
> >
> > 1. **Group answers and majority.**
> >    We sample $G$ rollouts, parse answers $a_i$, and compute the plurality answer $\hat{a}$. We then define
> >    $$
> >    M = \{ i : a_i = \hat{a} \} \quad \text{(majority set)}, \\
> >    R = \{ i : a_i \neq \hat{a} \} \quad \text{(residual set)},
> >    $$
> >    with majority share $\alpha = |M| / G$.
> >
> > 2. **Leave-one-out residual support.**
> >    For any candidate answer $b$ and any residual index $i \in R$, we define the leave-one-out residual support
> >    $$
> >    u^{(-i)}(b)
> >    = \frac{1}{|R|-1}
> >    \sum_{\substack{j \in R \\ j \neq i}} \mathbb{I}[a_j = b].
> >    $$
> >    For trajectory $i$, we set
> >    $$
> >    z_i =
> >    \begin{cases}
> >    u^{(-i)}(a_i), & i \in R, \\
> >    0, & i \in M.
> >    \end{cases}
> >    $$
> >
> > 3. **Residual mean $\bar{u}$.**
> >    The quantity $\bar{u}$ is simply the average of these $z_i$ over the residual set:
> >    $$
> >    \bar{u} = \frac{1}{|R|} \sum_{j \in R} z_j.
> >    $$
> >
> > 4. **ResZero reward.** The ResZero reward is then $$r_i^{\\text{ResZero}} = \\alpha \\cdot \\mathbb{I}[i \\in R]\\,(z_i - \\bar{u}) - c \\alpha \\cdot \\mathbb{I}[i \\in M] - \\gamma, \\qquad \\gamma = c \\alpha^2.$$
> >
> > Algorithmically, this is implemented by first counting the frequency of each distinct answer within $R$, computing $z_i$ from these counts, and then averaging to obtain $\bar{u}$. There is no additional approximation or sampling beyond the observed group of rollouts.
> >
> >
> > ### Response to Q2: What happens when all samples are different answers?
> >
> > In the extreme case where all $G$ answers are different, there is no “true” majority in the usual sense, but this configuration is fully covered by our general construction in Appendix A.2.
> >
> > 1. **Deterministic tie-breaking (edge case (i) in the Appendix A.2 footnote).**
> >    When multiple answers tie for the most frequent, including the all-distinct case, we select a single $\hat{a}$ by a fixed, deterministic rule (e.g., the earliest index). This ensures that $M = \{ i : a_i = \hat{a} \}$ and $R = \{ i : a_i \neq \hat{a} \}$ are uniquely defined, so the generic proof of the zero-mean property applies without any modification.
> >
> > 2. **Specialization of the general formula to the all-distinct configuration.**
> >    If every answer appears exactly once, then $|M| = 1$, $|R| = G - 1$, and $\alpha = |M|/G = 1/G$. For each residual trajectory $i \in R$ there are no other trajectories with the same answer, so $z_i = u^{(-i)}(a_i) = 0$ and hence $\bar{u} = 0$. Substituting these quantities into the expression from Appendix A.2 gives
> >    $$
> >    \sum_{i=1}^{G} r_i^{\text{ResZero}}
> >    = |M|(-c\alpha + \gamma) + |R|\gamma
> >    = -c\alpha|M| + G\gamma
> >    = 0,
> >    $$
> >    where the last equality follows directly from the choice $\gamma = c\alpha^2$ derived in Appendix A.2. Thus the **strict zero-mean property** continues to hold exactly in this degenerate setting; no additional case handling beyond the deterministic tie-breaking rule is required.
> >
> > 3. **Intuition.**
> >    Intuitively, when every trajectory proposes a unique answer, there is essentially no actionable consensus signal. In this case ResZero reduces to a very mild, zero-mean perturbation: one trajectory (the deterministically chosen “majority”) is slightly down-weighted and the remaining $G - 1$ trajectories are slightly up-weighted, with magnitude controlled by $\alpha = 1/G$. Because $\alpha$ is small in this regime, the overall effect on the policy is also small, and the fixed tie-breaking rule ensures that no single hypothesis is systematically favored or disfavored across batches.
> >
> > We have moved the footnote in Appendix A.2 into the body of Appendix A.2 to explicitly discuss the all-distinct case and clarify the behavior.
> >
> >
> > ### Response to Q3: Formal statement of the proposition in Appendix A.2
> >
> > We appreciate the request for a clearer statement. In the revised version, Appendix A.2 explicitly formulates the following proposition:
> >
> > > **Proposition (Zero-mean property of ResZero).**
> > > Consider any group of $G \ge 1$ trajectories, with majority set $M$, residual set $R$, and majority share $\alpha = |M| / G$. Let the ResZero reward $r_i^{\text{ResZero}}$ be defined as in Equation (5) with $\gamma = c \alpha^2$. Then, for all possible vote patterns (including ties and degenerate cases),
> > > $$
> > > \sum_{i=1}^{G} r_i^{\text{ResZero}} = 0.
> > > $$
> >
> > Appendix A.2 then proves this claim by:
> >
> > 1. Decomposing the sum over $M$ and $R$,
> > 2. Using the fact that $\sum_{i \in R} (z_i - \bar{u}) = 0$, and
> > 3. Showing that choosing $\gamma = c \alpha^2$ is exactly the condition that makes the combined sum zero.
> >
> > We have updated Appendix A.2 to clearly state the proposition in this form before presenting the proof.
> >
> > (Response continued in Part 3/3)

---

> > > ### Author Response · Authors · 2025-12-01
> > > **Response to Reviewer a7FE (Part 3/3)**
> > >
> > > (Continued from Part 2/3)
> > >
> > > ### Response to Q4: Zero-mean random reward baseline
> > >
> > > Thank you for this suggestion. In the revised version we have added a **random zero-mean reward** baseline as an additional fallback strategy when verification is inconclusive.
> > >
> > > Concretely, in this baseline we:
> > >
> > > 1. Draw i.i.d. random rewards $\tilde{r}_i$ for the $G$ trajectories (for example, from a symmetric distribution such as a standard normal).
> > >
> > > 2. Re-center them to enforce zero mean.
> > >
> > > This construction satisfies the same **zero-mean property** as ResZero, but it does **not** use any information about the vote structure (majority vs. residual, or relative popularity within residuals). As a result, it provides a noisy signal that is not aligned with which hypotheses are more promising.
> > >
> > > Empirically, across backbones and benchmarks, this random zero-mean baseline performs clearly worse than ResZero. This confirms that the specific structure of ResZero, which penalizes unverifiable consensus and amplifies relatively stronger residuals with magnitude controlled by $\alpha$, is crucial for obtaining the consistent gains.
> > > The detailed results for this baseline are provided in Table 3.
> > >
> > > ### Response to Q5: How are the confidence intervals in Table 1 and Table 2 (renumbered as Table 3 in the revised version) computed?
> > >
> > > In the original submission we reported mean $\pm$ standard error of the mean (SEM), not confidence intervals, which can be confusing. In the revised manuscript we now report **mean $\pm$ 95% confidence intervals (CIs)** for all entries in Tables 1 and 2 and clarify this explicitly in the text.
> > >
> > > The computation is as follows:
> > >
> > > * For each method and backbone, we run training with multiple random seeds.
> > > * For every seed, we compute
> > >   * per-benchmark metrics (for example, Avg@k), and
> > >   * the “Average” metric by averaging these per-benchmark metrics across datasets for that seed.
> > > * We then treat the per-seed values as i.i.d. samples and compute
> > >   * the sample mean $\hat{\mu}$, and
> > >   * the sample standard deviation $\hat{\sigma}$.
> > > * The reported 95% CI is
> > >   $$
> > >   \hat{\mu} \pm t_{0.975,\,n-1} \cdot \frac{\hat{\sigma}}{\sqrt{n}},
> > >   $$
> > >   where $n$ is the number of seeds and $t_{0.975,\,n-1}$ is the corresponding Student-$t$ quantile for $n - 1$ degrees of freedom.
> > >
> > >
> > > ### Response to Minor: “majority” vs. “plurality”
> > >
> > > We agree that, strictly speaking, “majority” typically means “$>50\%$,” whereas our usage aligns with the common “majority voting” terminology in the LLM literature, where it often denotes the most frequent (plurality) answer even when its share is below 50%.
> > > We have clarified this in the revised manuscript by explicitly stating this in a footnote in the **Introduction** section.

---

### Official Review · Reviewer_tvcp · 2025-11-01

**Soundness:** 3
**Presentation:** 2
**Contribution:** 3
**Rating:** 6
**Confidence:** 4

**Summary:**

This paper introduce a novel label-free RL framework (JURY-RL) that leverage model-based Learn Prover system to assign psudo-label for majority responses. By leverage strong LLMs to parse rollouts and run multiple verification trials, JURY-RL achieves better reward F1 than LLM-as-a-Judge baseline. Moreover, the author propose a simple ResZero reward to assign valid learning signal for prompts which have non-conclusive majority response and found this simple reward improves both pass@1 and pass@k performance.

**Strengths:**

1. Evaluation results shows that JURY-RL gives better results than other label-free RL method baselines.
2. Using a Lean verifier to assign psudo-label gives good reward quality.
3. The proposed simple ResZero reward effectively improves pass@k and is easy to use.

**Weaknesses:**

1. Evaluation: Avg@k results are required on benchmarks like AIME24, AIME25, the reliability is questionable. For example, at least 16 trials are required for AIME24.
2. Lack important baseline like TTRL (NeurIPS 25').
3. Efficiency: The author adopt a Pass@K verification setting, might causing 400 sec overhead, while the LLM-as-a-Judge baseline might only introduce 10 sec if also only evaluate the majority. Even when the LLM-as-a-Judge method rewards each response, 400 sec overhead budge can afford much more larger models like Qwen3-235B-A22B. The author might have to give more analysis on this efficiency comparison to demonstrate the effectiveness.

**Questions:**

1. In Table 2, it is confusing that the Proof-Gate + MV setting  performs much worse on Qwen models compared with Majority-Voting baseline.

---

> ### Author Response · Authors · 2025-12-01
> **Response to Reviewer tvcp (Part 1/2)**
>
> We thank the reviewer for the careful and constructive review, and for highlighting both the strengths of JURY-RL and several important points about evaluation, baselines, and efficiency. Below we address each concern in turn and describe the corresponding clarifications and revisions in the updated manuscript.
>
> ### 1. Response to W1: Evaluation on AIME24/25 and Avg@k reliability
>
> **Avg@k / Pass@k reporting and Reliability.**
> We completely agree that Avg@k/Pass@k is a more robust metric for benchmarks such as AIME. In the revised manuscript, we have **comprehensively updated our evaluation protocol** to report Avg@k results across almost all main experiments and ablation studies, instead of pass@1. As detailed in **Appendix F (Benchmark and Metric Details)**, we report avg@16 for AIME24/25 and using avg@4 or avg@8 for different size datasets like MATH500 and AMC. This ensures that the results reported in the main Table 1 and the Table 3 reflect rigorous estimates derived from consistent sampling settings.
>
> **Performance comparison.**
> Table 8 (formerly Table 7) reports these Pass@k results for all three backbones. The results demonstrate that JURY-RL achieves substantial improvements over the label-free baselines and remains competitive with the supervised GT-Reward baseline. For example:
>
> * For **Qwen2.5-7B**, JURY-RL improves the **pass@k** by **+10.05 points** over LLM-as-a-Judge (53.99 $\to$ 64.04) and by **+1.56 points** over GT-Reward.
> * For **Qwen3-1.7B**, JURY-RL improves the **pass@k** by **+9.06 points** over LLM-as-a-Judge (50.35 $\to$ 59.41) and by **+4.05 points** over GT-Reward (55.36 $\to$ 59.41).
>
> We believe these results effectively address the concern regarding the reliability of evaluation on AIME-style benchmarks. Furthermore, to provide a better measure of uncertainty, we have replaced “mean $\pm$ SEM” with **mean $\pm$ 95% confidence intervals** for all tables, allowing for a direct assessment of statistical significance.
>
> ### 2. Response to W2: Relation to TTRL and new experimental comparison
>
> We appreciate the suggestion to include TTRL [1] as a baseline. Conceptually, TTRL is a **test-time RL** method that constructs a pseudo-label from self-consistency among multiple rollouts on the test set and then adapts the model at test time using this signal.
>
> At a high level, this is very close to the **Majority-Voting baseline** that we already include and analyze extensively:
>
> * In our Majority-Voting baseline, for each problem we generate (G) rollouts.
> * A rollout receives reward 1 if its extracted answer matches the majority answer among the (G) rollouts, and 0 otherwise.
>
> This is essentially the same pseudo-labeling principle as TTRL’s reward: maximize agreement with the dominant answer, without checking correctness against ground truth.
>
> Our main experiments show that **JURY-RL consistently outperforms Majority-Voting** across benchmarks and backbones in the offline setting. These comparisons address the spirit of a TTRL-style baseline: they show that replacing purely self-consistent pseudo-labels with **formally verified rewards plus the ResZero fallback** gives a more reliable training signal.
>
> **New Experiment: Test-Time RL (Appendix: Test-Time Training Results).**
> To strictly verify this advantage in the reviewer's suggested setting, we have added a new section in the appendix where we apply JURY-RL **within the test-time training loop** and compare it against the standard TTRL (MV-based) baseline.
>
> Using the validation splits of AIME24, AIME25, AMC23, MATH500, and GSM8K as the environment, we ran test-time adaptation for both methods. The results (plotted in the newly added **Figure 6**) show:
>
> * **MV brittleness:** On difficult benchmarks like **AIME24 and AMC23**, the standard TTRL (MV reward) signal is brittle; performance initially rises but eventually degrades below the starting point. This confirms our analysis that spurious consensus can drive entropy collapse.
> * **JURY-RL stability:** In contrast, JURY-RL (using the Lean verifier + ResZero) maintains a stable learning curve and achieves higher final accuracy across benchmarks (e.g., attaining a stable plateau on AIME25 where MV fluctuates).
>
> TTRL provides a powerful framework for test-time adaptation, while JURY-RL provides a superior reward mechanism that can be plugged into that framework. By replacing the noisy majority-vote signal with our verifiable reward, we prevent the "collapse" phenomena observed in standard TTRL on hard tasks. We have updated the Appendix G.2 to explicitly discuss this relationship and include these new findings.
>
> [1] Zuo et al. (2025). *TTRL: Test-Time Reinforcement Learning.* NeurIPS 2025. arXiv:2504.16084.
>
> (Response continued in Part 2/2)

---

> > ### Author Response · Authors · 2025-12-01
> > **Response to Reviewer tvcp (Part 2/2)**
> >
> > (Continued from Part 1/2)
> >
> > ### 3. Response to W3: Efficiency comparison – Pass@k verification vs. LLM-as-a-Judge
> >
> > **Clarifying the timing basis and the "Early-Stopping" mechanism.**
> > Thank you for raising the efficiency concern. We believe part of the confusion stems from how the overheads were interpreted. In **Appendix C.4 Performance and Cost Analysis**, all timings are measured **per training step** (processing a batch of 128 questions), not per single problem. We wish to clarify two factors that keep our computational cost tractable:
> >
> > 1.  **Verification Scope:** For each question in the batch, we generate multiple rollouts but apply majority voting to select a *single* candidate answer. We verify *only* this consensus answer. Thus, there are 128 verification calls per step, not $128 \times G$.
> > 2.  **Early Stopping in Lean Verification:** A critical feature of our pipeline is the **early-stopping mechanism** inherent to the `pass@k` setting. In our framework, the system attempts to autoformalize and prove the statement up to $k$ times. However, **the process terminates immediately upon the first success.** We do not execute all $k$ trials if a valid proof is found earlier. Since we only need to verify the majority-voted candidates, which is very likely correct, the prover usually succeeds in the first few tries. This makes the actual compute time much smaller than the expected `pass@k` time.
> >
> > Under this configuration, we measure:
> > * The GT-Reward oracle (no external verification) at about **100 seconds per step**.
> > * JURY-RL (Lean verifier) adds **≈200 seconds per step** of overhead. This accounts for the average case where many proofs are found quickly via early stopping.
> > * LLM-as-a-Judge (Qwen-2.5-72B) adds **≈80 seconds per step**.
> >
> > Thus, the total step times are about **300 seconds** for JURY-RL versus **180 seconds** for LLM-as-a-Judge. The “400-second vs 10-second” figures mentioned in the review do not correspond to our reported settings.
> >
> > In the revised manuscript, we explicitly detail the **early-stopping mechanism** in **Appendix C.4** to explain why the `pass@k` verification remains efficient and transparently report these per-step timings.
> >
> >
> > ### 4. Response to Q1: Behavior of Proof-Gate + MV vs Majority-Voting in Table 2
> >
> > We appreciate the observation that in **Table 2 (renumbered as Table 3)** the “Proof-Gate + MV Reward” setting performs noticeably worse than the plain Majority-Voting baseline for Qwen models, which can seem counterintuitive at first. This behavior is actually consistent with our analysis of **reward hacking under self-consistency signals** and highlights why ResZero is necessary.
> >
> > Recall that:
> >
> > * **Majority-Voting (MV)** is a pure self-consistency signal. A rollout is rewarded if its answer matches the majority among the (G) rollouts, regardless of correctness.
> > * In **Proof-Gate + MV Reward**, we first apply the Lean-based proof gate.
> >
> >   * When the verifier succeeds (δ = 1), only trajectories supporting the verified answer are rewarded.
> >   * When the verifier fails (δ = 0), we fall back to MV Reward, that is, we reward the majority answer even though it has just failed verification.
> >
> > Our theoretical and empirical analysis shows that as an incorrect consensus strengthens (the majority vote becomes more dominant), MV tends to cause **reward hacking and entropy collapse**. Supporters of the dominant (possibly wrong) answer continue to receive positive signal, while alternative trajectories receive increasingly negative advantages, which drives the policy towards a single spurious mode and suppresses exploration.
> >
> > In this regime, if the majority answer is **wrong and unverifiable**, the Proof-Gate will almost always reject it (δ = 0), but the MV fallback will continue to **reward exactly this wrong majority**. This combination:
> >
> > 1. Fails to correct the spurious consensus, and
> > 2. Can accelerate collapse compared with plain MV, because the proof gate modulates gradients in a way that further discourages exploration away from the entrenched majority.
> >
> > Empirically, the revised Table 3 shows precisely this effect: for Qwen backbones, “Proof-Gate + MV Reward” achieves significantly lower Average accuracy than both Majority-Voting and “Proof-Gate + Zero Reward”, while **JURY-RL (Proof-Gate + ResZero)** consistently attains the best performance. In the revised manuscript, we expand the discussion in Section 5.3 to make this intuition explicit and to emphasize that “rewarding an unverified majority is deceptive.” ResZero is designed exactly to avoid this failure mode by providing a **zero-mean, variance-preserving fallback** instead of reinforcing an unverified consensus.
> >
> > ---
> >
> > We hope these clarifications on avg@k evaluation, the relation to TTRL, the efficiency comparison, and the behavior of Proof-Gate + MV address your concerns. We are grateful for your constructive comments, which have helped us improve both the experimental presentation and the explanation of our design choices.

---

### Official Review · Reviewer_sByP · 2025-11-04

**Soundness:** 3
**Presentation:** 3
**Contribution:** 2
**Rating:** 2
**Confidence:** 3

**Summary:**

This paper proposes a type of hybrid reward that combines a verifiable reward with a label-free heuristic reward that is used when the output fails to be verified. Verifiable rewards (from a formal theorem prover) are highly reliable, but sparse. They can also be expensive to compute. Label-free heuristics (e.g., majority voting or LLM-as-a-judge) can give denser rewards, but are more unreliable. The "Votes Propose, Proofs Dispose" method proposed here attempts to leverage both of these by (a) only verifying the majority voted response, and then (b) if that fails, adding a fallback mechanism for assigning nonzero reward to the non-majority (unverified) responses based on a label free heuristic.

**Strengths:**

The problem the paper addresses is significant, and a key practical challenge for LLM post-training. The proposed idea of using a cheap heuristic to filter candidates before applying an expensive verifier is natural. The fallback reward is also simple + intuitive (encouraging exploration of the next-highest-voted outcomes) that is zero-mean with non-zero variance.

**Weaknesses:**

The main idea of hybridizing sparse rewards with denser, heuristic signals is natural and not particularly new. For example, process rewards achieve a similar goal, but are not discussed or compared to in the paper. Hybrid rewards in particular have also been explored before in Huang et al 2025. The empirical results are also not entirely compelling: in Table 1 in particular almost all of the results appear have confidence intervals that substantially overlap with other methods.

[1] Huang et al 2025. Pitfalls of Rule- and Model-based Verifiers – A Case Study on Mathematical Reasoning. https://arxiv.org/pdf/2505.22203v1

**Questions:**

- Not all of the baselines appear to be defined. What is CoReward referring to?
- The motivation for only validating the majority-voted answer is computational efficiency, but it's not clear exactly what is given up by this tradeoff. It would be nice to see how this method scales, for example, given a higher density of applied verified rewards (e.g. to top-k).

**Details Of Ethics Concerns:**

I might be wrong, but it seems like the margins have been significantly altered on this submission?

---

> ### Author Response · Authors · 2025-12-01
> **Response to Reviewer sByP (Part 1/2)**
>
> We thank the reviewer for the feedback. However, we respectfully disagree with the assessment that our contribution is merely a generic application of hybrid rewards. Below, we clarify why reducing JURY-RL to "natural hybridization" overlooks the fundamental optimization challenges in **label-free RLVR** that prior methods fail to solve, and we present new statistical evidence (95% CIs and Pass@k) demonstrating robust significant gains.
>
> ### 1. Response to W1: Novelty and the "Label-Free" Challenge
>
> **Why JURY-RL is not just "hybridizing rewards".**
>
> The reviewer suggests that combining sparse and dense signals is "not particularly new," citing Process Rewards and Huang et al. (2025)[1]. This comparison overlooks the critical constraint of our setting: **Label-Free RLVR with a Formal Verifier**.
>
> * **Process Rewards (PRMs) are not Label-Free:** Standard process reward methods rely on training PRMs using human annotations or distilling strong teacher models. This reintroduces the very bottleneck (supervision cost) we aim to eliminate. JURY-RL operates **without** training separate reward models or requiring step-level labels.
> * **Existing Hybrids Cause Collapse:** Naive hybridization (e.g., using majority voting or LLM-judges as fillers) in a label-free setting is mathematically unstable. As shown in our analysis (Section 5.3 & App. A), these methods actively drive the policy toward **mode collapse** (spurious consensus) because they reinforce unverified confidence.
> * **ResZero is a Theoretical Correction, Not Just a Heuristic:** Our contribution is not simply "adding a fallback." We derive **ResZero** specifically to solve the gradient collapse problem in GRPO. By constructing a **zero-mean, variance-preserving** signal, ResZero mathematically prevents the "rich-get-richer" feedback loop of standard heuristics. This is a novel optimization contribution tailored for the binary nature of formal verifiers, which prior hybrid works do not propose.
>
> **Differentiation from Huang et al. (2025).**
>
> Huang et al. (2025) provide a *diagnosis* of verifier pitfalls; JURY-RL provides a *solution*.
> * They show that hybrid verifiers (combining rule-based and model-based) are vulnerable to reward hacking.
> * We explicitly **reject** the use of learned verifiers/judges to avoid this hacking. Instead, we propose a novel architecture ("Votes Propose, Proofs Dispose") that uses the *empirical distribution of the policy itself* (via ResZero) to guide learning when the formal prover is silent. This approach is orthogonal to, and more robust than, the hybrid verifiers analyzed in Huang et al.
>
> [1] Huang et al. (2025). *Pitfalls of Rule- and Model-based Verifiers – A Case Study on Mathematical Reasoning*. [https://arxiv.org/pdf/2505.22203v1](https://arxiv.org/pdf/2505.22203v1)
>
> ### 2. Response to W1: Empirical Significance (95% Confidence Intervals)
>
> The reviewer noted that results appeared to have overlapping intervals. We clarify that our initial submission reported **Standard Error of the Mean (SEM)**, which is often much narrower than Confidence Intervals (CIs), leading to potential misinterpretation.
>
> To rigorously address this, we have **re-run our analysis to report 95% Confidence Intervals** across multiple seeds (updated in Table 3). The new analysis confirms:
> * **Statistically Significant Gains:** JURY-RL's improvements are statistically significant at the 95% level against all label-free baselines (Majority Voting, LLM-as-a-Judge, Entropy, Self-Certainty).
> * **Pass@k Dominance (New Table 2):** In realistic reasoning scenarios (Pass@k), the gap is even more pronounced. For example, on Qwen2.5-7B, JURY-RL outperforms the LLM-as-a-Judge baseline by **+10.05%** in Pass@k. This is a massive margin that far exceeds any statistical noise.
>
> (Response continued in Part 2/2)

---

> > ### Author Response · Authors · 2025-12-01
> > **Response to Reviewer sByP (Part 2/2)**
> >
> > (Continued from Part 1/2)
> > ### 3. Response to Q1: The CoReward Baseline
> >
> > We have clarified the definition of CoReward [2] in the revision. It is a self-supervised method leveraging contrastive agreement. Our results show JURY-RL outperforms CoReward because CoReward still relies on pseudo-labels that can drift, whereas JURY-RL anchors the reward scale to grounded formal verification.
> >
> > [2] Zhang et al. (2025). *Co-Reward: Self-supervised Reinforcement Learning for Large Language Model Reasoning via Contrastive Agreement*. [http://arxiv.org/abs/2508.00410](http://arxiv.org/abs/2508.00410)
> >
> > ### 4. Response to Q2: Majority-Only vs. Top-k Verification (Cost-Benefit)
> >
> > The reviewer asks what is given up by verifying only the majority answer. We added a direct ablation (Figure 8 in Appendix G.3) comparing Top-1 (Majority) vs. Top-2 and Top-3 verification.
> > * **Result:** The learning curves are nearly indistinguishable.
> > * **Conclusion:** In the RLVR setting, the "learning signal" is dominated by the most probable answer. Verifying additional candidates increases compute cost linearly without yielding meaningful performance gains. Thus, "Votes Propose, Proofs Dispose" (Top-1 verification) is not just a compromise, but the **pareto-optimal** strategy for scaling formal verification in RL.
> >
> > ### 5. Ethics & Formatting
> >
> > We apologize for the margin oversight caused by a compilation error. We have strictly reverted to the official ICLR formatting. Regarding the ethics flag: our work uses standard open datasets and open-source models; no new hazards are introduced.
> >
> > ---
> >
> > In conclusion, JURY-RL is not a simple hybrid heuristic but a theoretical solution to optimization collapse in label-free RLVR. Our revised analysis with 95% confidence intervals and decisive Pass@k gains confirms statistically significant improvements. We thus provide a robust and scalable pathway for reasoning alignment without human labels.

---

### Author Response · Authors · 2025-12-01
**General Response**

We thank all reviewers for their insightful feedback and constructive suggestions. We are encouraged that the reviewers recognize the **significance of label-free RLVR** (sByP, a7FE), the **mathematical soundness** of our approach (a7FE, kMKV), and the **comprehensive experimental validation** (kMKV, a7FE, tvcp).

In this revision, we have significantly strengthened the manuscript to address all reviewer questions. We wish to draw the Area Chair’s attention to **four critical updates** that demonstrate the distinct advantages of JURY-RL:

### 1. Highlighting Diversity & Pass@k Superiority (Section 5.3 Reorganization)
To better showcase our most significant empirical gain, **we have moved the "Does JURY-RL Enhance Diversity?" analysis to the beginning of Section 5.3.**
While JURY-RL matches the strong supervised GT-Reward baseline in average accuracy (`avg@k`), it **significantly outperforms** all baselines in `pass@k` (latent reasoning capability). As detailed in the new **Table 2** (main text) and **Table 8** (Appendix):
* **Qwen2.5-7B:** JURY-RL improves `pass@k` by **+10.05 points** over LLM-as-a-Judge and **+1.56 points** over the GT-Reward oracle.
* **Qwen3-1.7B:** JURY-RL improves `pass@k` by **+9.06 points** over LLM-as-a-Judge and **+4.05 points** over GT-Reward.
This confirms that JURY-RL prevents the mode collapse common in label-free methods (and even supervised learning), learning diverse, generalizable reasoning paths.

### 2. Comprehensive Pass@k Ablations (New Table 9)
We have added **Table 9**, reporting `pass@k` results for all ablation settings. This provides conclusive evidence that the **ResZero** reward is the specific driver of our performance gains. ResZero consistently achieves the highest `pass@k` and `avg@k` compared to "Proof-Gate + Zero" and "Proof-Gate + MV", confirming that our zero-mean, variance-preserving fallback is essential for optimization stability.

### 3. Addressing Reviewer kMKV’s Conditional Score Increase
Reviewer **kMKV** explicitly stated: *"I will increase the score if all the questions are well-justified."*
We have fully addressed kMKV’s concerns regarding cost-benefit and reliability:
* **Efficiency:** We added **Figure 8**, showing that verifying only the **Top-1** candidate yields the same performance as Top-3, minimizing compute cost. We also clarified that **early stopping** significantly reduces verification time compared to LLM-as-a-Judge (once model size is accounted for).
* **Reliability:** We added **Appendix C.3**, evaluating each pipeline module in isolation (Autoformalizer: 95.4% coverage; Consistency Checker: 91.1% F1; Prover: 0% False Positives).
We believe these new data points satisfy the condition for the score increase.

### 4. Robustness Checks: New Baselines & 95% CIs
* **New Test-Time Training (TTRL) Baseline:** We added **Appendix G.2** and **Figure 6**, comparing JURY-RL against a TTRL (Majority Voting) baseline. The results show that while standard MV is brittle and collapses on hard tasks (AIME/AMC) during test-time adaptation, JURY-RL maintains a stable, improving learning curve.
* **Statistical Significance:** We have re-computed all main results using **95% Confidence Intervals** (rather than SEM) to demonstrate that our improvements are statistically significant.

**Summary of Revisions:**
* **New Experiments:** Top-k verification ablation (Fig 8), isolated verifier component analysis (App C.3), TTRL comparison (Fig 6), Random-Zero baseline (Table 3).
* **Metric Updates:** Reported `avg@k` in main tables; updated to 95% CIs.
* **Clarifications:** Defined CoReward and ResZero formalisms explicitly; corrected formatting margins.

 We have updated the manuscript to reflect these clarifications, with major revisions highlighted in blue. We believe JURY-RL now stands as a rigorously validated, scalable solution for reasoning alignment that effectively bridges the gap between label-free learning and formal verification.

---

### Meta-Review · Area_Chair_DZp8 · 2025-12-08

**Summary:**

This paper focuses on the use of a Lean prover to propagate reward during learning (using the frameworkof verifiable rewards). The reviewers had some hesitations about the paper, there were especially concerns about the evaluation, which I am not sure have been properly addressed. In addition, it seems like the paper was not formatted correctly originally (margins wider?) - which could potentially mean it should have been desk rejected (depending on the exact ICLR policy).

**Reviewer Concerns:**

-

**Reviewer Scores:**

I cannot predict that.

---

### Decision · Program_Chairs · 2026-01-26

Reject